# KL-Regularized RLHF with Multiple Reference Models: Exact Solutions and Sample Complexity

**Gholamali Aminian**\*
The Alan Turing Institute
London, UK
gaminian@turing.ac.uk

**Amir R. Asadi**
Statistical Laboratory
University of Cambridge, UK
asadi@statslab.cam.ac.uk

**Idan Shenfeld**
Massachusetts Institute of Technology
USA
idanshen@mit.edu

**Youssef Mroueh**
IBM Research
USA
mroueh@us.ibm.com

## Abstract

Recent methods for aligning large language models (LLMs) with human feedback predominantly rely on a single reference model, which limits diversity, model overfitting, and underutilizes the wide range of available pre-trained models. Incorporating multiple reference models has the potential to address these limitations by broadening perspectives, reducing bias, and leveraging the strengths of diverse open-source LLMs. However, integrating multiple reference models into reinforcement learning with human feedback (RLHF) frameworks poses significant theoretical challenges, where achieving exact solutions has remained an open problem. This paper presents the first *exact solution* to the multiple reference model problem in reverse KL-regularized RLHF. We introduce a comprehensive theoretical framework that includes rigorous statistical analysis and provides sample complexity guarantees. Additionally, we extend our analysis to forward KL-regularized RLHF, offering new insights into sample complexity requirements in multiple reference scenarios. Our contributions lay the foundation for more advanced and adaptable LLM alignment techniques, enabling the effective use of multiple reference models. This work paves the way for developing alignment frameworks that are both theoretically sound and better suited to the challenges of modern AI ecosystems.

## 1 Introduction

Large language models (LLMs) have revolutionized natural language processing (NLP) by demonstrating remarkable capabilities in understanding and generating human language. Powered by vast datasets and advanced neural architectures, these models have set new benchmarks across various NLP tasks, including machine translation and conversational agents. Despite these advancements, aligning LLMs with human values and preferences remains a critical challenge. Such misalignment can lead to undesirable behaviors, including the generation of biased or inappropriate content, which undermines the reliability and safety of these models [Gehman et al., 2020].

Reinforcement Learning from Human Feedback (RLHF) has emerged as a pivotal framework for addressing alignment challenges in LLMs. By fine-tuning LLMs based on human feedback, RLHF steers models towards more human-aligned behaviors, enhancing truthfulness, helpfulness, and

---

\*Corresponding author.

39th Conference on Neural Information Processing Systems (NeurIPS 2025).

harmlessness while maintaining their ability to generate accurate and high-probability outputs [Wirth et al., 2017, Christiano et al., 2017]. In RLHF, reward-based methods use a trained reward model to evaluate (prompt, response) pairs. These methods treat the language model as a policy that takes a prompt $x$ and generates a response $y$ conditioned on $x$, optimizing this policy to generate responses with maximum reward. Typically, a reference policy (usually the pretrained model before fine-tuning) is used as a baseline to regularize training, preventing excessive deviation from the original behavior.

An inherent limitation of most works on LLM alignment is their reliance on a *single reference model* [Wang et al., 2024b]. First, this restricts the diversity of linguistic patterns and inductive biases available during training. In that, it is over restrictive - potentially leading to a model that inherits the limitations or cultural biases of a single pretrained source. Second, such an approach is inefficient in utilizing the wealth of pre-trained models available in modern AI ecosystems, which excel in different domains and capture unique nuances, leaving valuable collective intelligence untapped. Therefore, incorporating multiple LLMs as reference models produces a model that reflects the characteristics of all reference models while satisfying human preferences. This approach is particularly relevant as the open-source community continues to release diverse pre-trained and fine-tuned LLMs of varying scales, trained on a wide range of datasets [Jiang et al., 2023, Penedo et al., 2023].

A solution is to extend the RLHF training to utilize *multiple reference models*. While RLHF with multiple reference models has demonstrated practical utility [Le et al., 2024], its theoretical underpinnings remain largely unexplored. A critical gap in current understanding is the lack of an exact solution for reverse KL-regularized RLHF when incorporating multiple reference models. This theoretical limitation has prevented the study of sample complexity of bounds on both optimality and sub-optimality gaps in the reverse KL-regularized framework. Addressing this problem is crucial for advancing the alignment of LLMs with human preferences in increasingly complex and diverse settings.

In this work, we provide the solutions for RLHF with multiple reference models when regularized via Reverse KL divergence (RKL) or forward KL divergence (FKL). In addition, we provide a statistical analysis of these scenarios. Our main contributions are as follows:

- We propose a comprehensive mathematical framework for reverse KL-regularized RLHF with multiple reference models and provide the exact solution for this problem and calculate the maximum objective value.
- We provide theoretical guarantees for the proposed multiple reference models scenario under reverse KL-regularization. In particular, we study the sample complexity[2] of reverse KL-regularized RLHF under multiple reference models.
- We also study the multiple reference models scenario under forward KL-regularized RLHF and analyze its sample complexity.

## 2   Related Works

**Multiple References:** Inspired by model soups [Wortsman et al., 2022], Chegini et al. [2024] propose a reference soup policy, achieved by averaging two independently trained supervised fine-tuned models, including the reference model. However, their approach lacks theoretical guarantees, particularly regarding its applicability to alignment tasks. More recently, Le et al. [2024] introduced the concept of multiple reference models for alignment. Due to the challenges in deriving a closed-form solution for the RLHF objective under multiple referencing constraints, the authors proposed a lower-bound approximation. In this work, we address this gap by deriving the closed-form solution for the multiple reference model scenario under reverse KL-regularization.

**Theoretical Foundation of RLHF:** Several works have studied the theoretical underpinnings of reverse KL-regularized RLHF, particularly in terms of sample complexity [Zhao et al., 2024, Xiong et al., 2024, Song et al., 2024, Zhan et al., 2023, Ye et al., 2024]. Among these, Zhao et al. [2024] analyze reverse KL-regularized RLHF, demonstrating the effect of reverse KL-regularization and establishing an upper bound on sub-optimality gap with $O(1/n)$ sample complexity (convergence rate) where $n$ represents the size of preference dataset. More detailed comparison with these works is provided in Section 7. However, to the best of our knowledge, the RLHF framework incorporating multiple reference models has not yet been studied.

---

[2]The sample complexity provides insight into how quickly bounds converge as the dataset size increases.

**Forward KL-regularization and Alignment:** The forward KL-regularization for Direct Preference Optimization (DPO) proposed by Wang et al. [2024a]. The application of forward KL-regularization for alignment from demonstrations is shown in [Sun and van der Schaar, 2024]. The forward KL-regularization in stochastic decision problems is also studied by Cohen [2017]. To the best of our knowledge, the forward KL-regularized RLHF is not studied from a theoretical perspective.

# 3 Preliminaries

**Notations:** Upper-case letters denote random variables (e.g., $Z$), lower-case letters denote the realizations of random variables (e.g., $z$), and calligraphic letters denote sets (e.g., $\mathcal{Z}$). All logarithms are in the natural base. The set of probability distributions (measures) over a space $\mathcal{X}$ with finite variance is denoted by $\mathcal{P}(\mathcal{X})$. The KL-divergence between two probability distributions on $\mathbb{R}^d$ with densities $p(x)$ and $q(x)$, so that $q(x) > 0$ when $p(x) > 0$, is $\mathrm{KL}(p\|q) := \int_{\mathbb{R}^d} p(x) \log(p(x)/q(x)) \mathrm{d}x$ (with $0/0 := 0$). The entropy of a distribution $p(x)$ is denoted by $H(p) = -\int_{\mathbb{R}^d} p(x) \log(p(x))$.

We define the Escort and Generalized Escort distributions [Bercher, 2012] (a.k.a. normalized geometric transformation).

**Definition 3.1** (Escort and Generalized Escort Distributions)**.** *Given a discrete probability measure $P$ defined on a set $\mathcal{A}$, and any $\lambda \geq 0$, we define the escort distribution $(P)^\lambda$ for all $a \in \mathcal{A}$ as*

$$(P)^\lambda(a) := \frac{(P(a))^\lambda}{\sum_{x \in \mathcal{A}} (P(x))^\lambda}.$$

*Given two discrete probability measures $P$ and $Q$ defined on a set $\mathcal{A}$, and any $\lambda \in [0,1]$, we define the generalized escort distribution $(P,Q)^\lambda$ as the following tilted distribution:*

$$(P,Q)^\lambda(a) := \frac{P^\lambda(a) Q^{1-\lambda}(a)}{\sum_{x \in \mathcal{A}} P^\lambda(x) Q^{1-\lambda}(x)}.$$

Next, we introduce the functional derivative, see Cardaliaguet et al. [2019].

**Definition 3.2.** *[Cardaliaguet et al., 2019] A functional $U : \mathcal{P}(\mathbb{R}^n) \to \mathbb{R}$ admits a functional derivative if there is a map $\frac{\delta U}{\delta m} : \mathcal{P}(\mathbb{R}^n) \times \mathbb{R}^n \to \mathbb{R}$ which is continuous on $\mathcal{P}(\mathbb{R}^n)$ and, for all $m, m' \in \mathcal{P}(\mathbb{R}^n)$, it holds that*

$$U(m') - U(m) = \int_0^1 \int_{\mathbb{R}^n} \frac{\delta U}{\delta m}(m_\lambda, a) (m' - m)(da) \, \mathrm{d}\lambda,$$

*where $m_\lambda = m + \lambda(m' - m)$.*

We also define the sensitivity of a policy $\pi_r(y|x)$, which is a function of reward function $r(x,y)$, with respect to the reward function as

$$\frac{\partial \pi}{\partial r}(r) := \lim_{\Delta r \to 0} \frac{\pi_r(y|x) - \pi_{r+\Delta r}(y|x)}{\Delta r}. \tag{1}$$

# 4 Problem Formulation

Following prior works [Ye et al., 2024, Zhao et al., 2024], we consider the problem of aligning a policy $\pi$ with human preferences. Given an input (prompt) $x \in \mathcal{X}$ which is samples from $\rho(x)$, is the finite space of input texts, the policy $\pi \in \Pi$, where $\Pi$ is the set of policies, models a conditional probability distribution $\pi(y|x)$ over the finite space of output texts $y \in \mathcal{Y}$. From a given $\pi$ and $x$, we can sample an output (response) $y \sim \pi(\cdot|x)$.

**Preference Dataset:** Preference data is generated by sampling two outputs $(y, y')|x$ from $\pi_{\mathrm{ref}}$ as the reference policy (model), and presenting them to an agent, typically a human, for rating to indicate which one is preferred. For example, $y \succ y'$ denotes that $y$ is preferred to $y'$. A preference dataset is then denoted as $D = \{y_i^w, y_i^l, x^i\}_{i=1}^n$, where $n$ is the number of data points, $y_w$ and $y_l$ denote the preferred (chosen) and dispreferred (rejected) outputs, respectively.

We assume that there exists a true model of the agent's preference $p^*(y \succ y'|x)$, which assigns the probability of $y$ being preferred to $y'$ given $x$ based on the latent reward model which is unknown.

## 4.1 RLHF from One Reference Model

Using the dataset $\mathcal{D}$, our goal is to find a policy $\pi$ that maximizes the expected preference while being close to a reference policy $\pi_{\text{ref}}$. In this approach, Bradley–Terry model Bradley and Terry [1952] is employed as the preference model, $p(y \succ y'|x) = \sigma(r_\theta(x, y) - r_\theta(x, y'))$,

where $\sigma$ denotes the sigmoid function and $r_\theta : \mathcal{X} \times \mathcal{Y} \rightarrow \mathbb{R}$ is a reward model parameterized by $\theta$, which assigns a scalar score to indicate the suitability of output $y$ for input $x$. In Christiano et al. [2017], the reward model is trained on $D$ to maximize the log-likelihood (MLE) estimator:

$$\mathcal{L}_R(\theta, D) = \sum_{i=1}^{n} \frac{1}{n} \log \sigma(r_\theta(x_i, y_w^i) - r_\theta(x_i, y_l^i)). \tag{2}$$

Given a trained reward model $r_{\widehat{\theta}}(x, y)$ where $\widehat{\theta} = \arg\max_{\theta \in \Theta} \mathcal{L}_R(\theta, D)$, we can consider the regularized optimization objective which is regularized via reverse KL-regularized or forward KL-regularized.

**Reverse KL-regularized RLHF:** A crucial component of RLHF is the use of a reference model to compute a Reverse Kullback-Leibler (KL) divergence penalty. This penalty ensures that the process does not deviate excessively from the original model, mitigating the risk of generating nonsensical responses [Ziegler et al., 2019]. The reverse KL-regularized optimization objective for $(\gamma > 0)$ can represented as:

$$\max_{\pi} \mathbb{E}_{Y \sim \pi(\cdot|x)}[r_{\widehat{\theta}}(x, Y)] - \frac{1}{\gamma}\text{KL}(\pi(\cdot|x)\|\pi_{\text{ref}}(\cdot|x)), \tag{3}$$

Note that the solution of (3) is,

$$\pi_{\widehat{\theta}}^{\gamma}(y|x) := \frac{\pi_{\text{ref}}(y|x) \exp(\gamma r_{\widehat{\theta}}(x, y))}{Z(x)}, \tag{4}$$

where $Z(x) = \mathbb{E}_{Y \sim \pi_{\text{ref}}(\cdot|x)}[\exp(\gamma r_{\widehat{\theta}}(x, Y))]$ is the normalization factor. Similarly, we can define $\pi_{\theta^\star}^{\gamma}(y|x)$ using $r_{\theta^\star}(x, y)$ instead of $r_{\widehat{\theta}}(x, y)$ in (4). This RLHF objective is employed to train LLMs such as Instruct-GPT Ouyang et al. [2022] using PPO Schulman et al. [2017].

We define $J(\pi_{\theta^\star}(\cdot|x)) = \mathbb{E}_{Y \sim \pi_{\theta^\star}(\cdot|x)}[r_{\theta^\star}(x, Y)]$ (a.k.a. value function[3]) and provide an upper bound on optimal gap,

$$\mathcal{J}(\pi_{\theta^\star}^{\gamma}(\cdot|x), \pi_{\widehat{\theta}}^{\gamma}(\cdot|x)) := J(\pi_{\theta^\star}^{\gamma}(\cdot|x)) - J(\pi_{\widehat{\theta}}^{\gamma}(\cdot|x)). \tag{5}$$

Furthermore, inspired by [Song et al., 2024, Zhao et al., 2024], we consider the following RLHF objective function based on the true reward function,

$$J_\gamma(\pi_{\text{ref}}(\cdot|x), \pi_\theta(\cdot|x)) := \mathbb{E}_{Y \sim \pi_\theta(\cdot|x)}[r_{\theta^\star}(Y, x)] - \frac{1}{\gamma}\text{KL}(\pi_\theta(\cdot|x)\|\pi_{\text{ref}}(\cdot|x)). \tag{6}$$

As studied by Zhao et al. [2024], Song et al. [2024], Zhan et al. [2023], we also aim to study the following sub-optimality gap,

$$\mathcal{J}^\gamma(\pi_{\theta^\star}^{\gamma}(\cdot|x), \pi_{\widehat{\theta}}^{\gamma}(\cdot|x)) := J_\gamma(\pi_{\text{ref}}(\cdot|x), \pi_{\theta^\star}^{\gamma}(\cdot|x)) - J_\gamma(\pi_{\text{ref}}(\cdot|x), \pi_{\widehat{\theta}}^{\gamma}(\cdot|x)). \tag{7}$$

**Forward KL-regularized RLHF:** Inspired by [Wang et al., 2024a], we can consider the forward KL-regularized optimization objective as,

$$\max_{\pi} \mathbb{E}_{Y \sim \pi(\cdot|x)}[r_{\widehat{\theta}}(x, Y)] - \frac{1}{\gamma}\text{KL}(\pi_{\text{ref}}(\cdot|x)\|\pi(\cdot|x)), \tag{8}$$

As discussed in [Wang et al., 2024a], this optimization problem has an implicit solution given by:

$$\tilde{\pi}_{\widehat{\theta}}^{\gamma}(y|x) := \frac{\pi_{\text{ref}}(y|x)}{\gamma(\tilde{Z}_{\widehat{\theta}}(x) - r_{\widehat{\theta}}(x, y))} \tag{9}$$

---

[3]We can also consider $\mathbb{E}_{X \sim \rho(\cdot)}[J(\pi(\cdot|X))]$. All of our results also holds for expected version of value function.

where $\tilde{Z}_{\widehat{\theta}}(x)$ is normalization constant ensuring that $\int_{\mathcal{Y}} \pi_\theta^\gamma(y|x)dy = 1$. Some properties of $\tilde{Z}_{\widehat{\theta}}(x)$ are discussed in App. E.

Similar to (5) and (7), for forward KL-regularized RLHF, we can define,

$$\tilde{J}_\gamma(\pi_{\mathrm{ref}}(\cdot|x), \pi_\theta(\cdot|x)) := \mathbb{E}_{Y \sim \pi_\theta(\cdot|x)}[r_{\theta^\star}(Y, x)] - \frac{1}{\gamma}\mathrm{KL}(\pi_{\mathrm{ref}}(\cdot|x) \| \pi_\theta(\cdot|x)), \tag{10}$$

$$\widetilde{\mathcal{J}}^\gamma(\tilde{\pi}_{\theta^\star}^\gamma(\cdot|x), \tilde{\pi}_{\widehat{\theta}}^\gamma(\cdot|x)) := \tilde{J}_\gamma(\pi_{\mathrm{ref}}(\cdot|x), \tilde{\pi}_{\theta^\star}^\gamma(\cdot|x)) - \tilde{J}_\gamma(\pi_{\mathrm{ref}}(\cdot|x), \tilde{\pi}_{\widehat{\theta}}^\gamma(\cdot|x)). \tag{11}$$

### 4.2 Assumptions

For our analysis, the following assumptions are needed.

**Assumption 4.1** (Bounded Reward). *We assume that the true and parametrized reward functions, $r_{\theta^\star}(x, y)$ and $r_{\widehat{\theta}}(x, y)$, are non-negative functions and bounded by $R_{\max}$.*

**Assumption 4.2** (Finite Class). *We assume that the reward function class, $\mathcal{R}$, is finite, $|\mathcal{R}| < \infty$.*

The assumption of bounded rewards (Assumption 4.1) and Finite class (Assumption 4.2) are common in the literature [Song et al., 2024, Zhan et al., 2023, Zhao et al., 2024, Chang et al., 2024, Xiong et al., 2024]. More discussion regarding these assumptions are provided in App. C.

Coverage conditions play a fundamental role in understanding the theoretical guarantees of RLHF algorithms. We first introduce the most stringent coverage requirement, known as global coverage [Munos and Szepesvári, 2008]:

**Assumption 4.3** (Global Coverage). *For all policies $\pi$, we require $\max_{x,y:\rho(x)>0} \frac{\pi(y|x)}{\widehat{\pi}_{\mathrm{ref}}(y|x)} \leq C_{\mathrm{GC}}$, where $\widehat{\pi}_{\mathrm{ref}}$ denotes the reference model and $C_{\mathrm{GC}} \in \mathbb{R}^+$ is a finite constant.*

A key implication of Assumption 4.3 is that it requires substantial coverage: specifically, for any prompt $x$ and token sequence $y$ in the support of $\rho$, we must have $\widehat{\pi}_{\mathrm{ref}}(y|x) \geq \frac{1}{C_{\mathrm{GC}}}$.

While global coverage has been extensively studied in the offline RL literature [Uehara and Sun, 2021, Zhan et al., 2022], it imposes strong requirements that may be unnecessarily restrictive for RLHF. A key insight from recent work [Zhao et al., 2024, Song et al., 2024] is that RLHF algorithms inherently employ reverse KL-regularization, which ensures learned policies remain within a neighborhood of the reference model. This observation motivates a more refined coverage condition:

**Assumption 4.4** (Local Reverse KL-ball Coverage). *Consider $\varepsilon_{\mathrm{rkl}} < \infty$ and any policy $\pi$ satisfying $\mathbb{E}_{x\sim\rho}[\mathrm{KL}(\pi(\cdot|x) \| \widehat{\pi}_{\mathrm{ref}}(\cdot|x))] \leq \varepsilon_{\mathrm{rkl}}$, we require $\max_{x,y:\rho(x)>0} \frac{\pi(y|x)}{\widehat{\pi}_{\mathrm{ref}}(y|x)} \leq C_{\varepsilon_{\mathrm{rkl}}}$, where $C_{\varepsilon_{\mathrm{rkl}}} \in \mathbb{R}^+$ depends on the KL threshold $\varepsilon_{\mathrm{rkl}}$.*

Similar to Assumption 4.4, we consider the forward KL-ball coverage assumption.

**Assumption 4.5** (Local Forward KL-ball Coverage). *Consider $\varepsilon_{\mathrm{fkl}} < \infty$ and any policy $\pi$ satisfying $\mathbb{E}_{x\sim\rho}[\mathrm{KL}(\widehat{\pi}_{\mathrm{ref}}(\cdot|x) \| \pi(\cdot|x))] \leq \varepsilon_{\mathrm{fkl}}$, we require $\max_{x,y:\rho(x)>0} \frac{\pi(y|x)}{\widehat{\pi}_{\mathrm{ref}}(y|x)} \leq C_{\varepsilon_{\mathrm{fkl}}}$, where $C_{\varepsilon_{\mathrm{fkl}}} \in \mathbb{R}^+$ depends on the KL threshold $\varepsilon_{\mathrm{fkl}}$.*

The local reverse or forward KL-ball coverage condition offers several advantages. Focusing only on policies within a reverse KL-ball of the reference model provides sharper theoretical guarantees while imposing weaker requirements. This localization aligns naturally with RLHF algorithms, which explicitly constrain the learned policy's divergence from the reference model. For any fixed reference model $\pi_{\mathrm{ref}}$, the reverse or forward KL local coverage constant is always bounded by the global coverage constant: $\max(C_{\varepsilon_{\mathrm{rkl}}}, C_{\varepsilon_{\mathrm{fkl}}}) \leq C_{\mathrm{GC}}$. This follows from the fact that KL-constrained policies form a subset of all possible policies.

## 5 RLHF from Multiple Reference Models via Reverse KL divergence

In this section, inspired by Le et al. [2024], we are focused on situations involving $K$ reference policies $\left\{\pi_{\mathrm{ref},i}\right\}_{i=1}^K$ where the latent reward model among all reference policies is the same. All proof details are deferred to Appendix D.

## 5.1 Exact Solution of RLHF under multiple reference models via RKL

Inspired by [Le et al., 2024], our objective can be formulated as a multiple reference models RLHF objective,

$$\max_{\pi} \mathbb{E}_{Y \sim \pi(\cdot|x)}\big[r(x, Y)\big] - \frac{1}{\gamma}\Big( \sum_{i=1}^{K} \alpha_i \mathrm{KL}\big(\pi(\cdot|x)\|\pi_{\mathrm{ref},i}(\cdot|x)\big)\Big), \tag{12}$$

where $\alpha_i$ are weighting coefficients for each reference policy and $\sum_{i=1}^{K} \alpha_i = 1$. This objective was explored in previous studies, leading to enhancements in pure RL problems Le et al. [2022].

However, addressing this optimization problem in LLMs through reward learning and RL finetuning pose similar challenges to (3). Our goal is to derive a closed-form solution for the multi-reference RLHF objective in (12). Note that in [Le et al., 2024, Proposition 1], a lower bound on RLHF objective in (12) is proposed, and the solution for this surrogate objective function is derived as follows,

$$\pi_{\mathrm{L}}\big(y|x\big) = \frac{\widetilde{\pi}_{\mathrm{ref}}\big(y|x\big)}{\widehat{Z}_1(x)} \exp\Big(\gamma r\big(x, y\big)\Big), \tag{13}$$

where $\widetilde{\pi}_{\mathrm{ref}}(y|x) = \Big( \sum_{i=1}^{K} \frac{\alpha_i}{\pi_{\mathrm{ref},i}(y|x)}\Big)^{-1}$ and $\widehat{Z}_1(x) = \sum_y \widetilde{\pi}_{\mathrm{ref}}(y|x) \exp\big(\gamma r(x, y)\big)$.

In contrast, in the following theorem, we provide the *exact* solution of the objective function for the multiple reference model (12).

**Theorem 5.1.** *Consider the following objective function for RLHF with multiple reference models,*

$$\max_{\pi}\Big\{ \mathbb{E}_{Y \sim \pi(\cdot|x)}\big[r_{\theta^\star}\big(x, Y\big)\big] - \frac{1}{\gamma}\Big( \sum_{i=1}^{K} \alpha_i \mathrm{KL}\big(\pi(\cdot|x)\|\pi_{\mathrm{ref},i}(\cdot|x)\big)\Big)\Big\},$$

*where $\sum_{i=1}^{K} \alpha_i = 1$ and $\alpha_i \in (0, 1)$ for $i \in [K]$. Then, the exact solution of the multiple reference model objective function for RLHF is,*

$$\pi_{\theta^\star}^{\gamma}\big(y|x\big) = \frac{\widehat{\pi}_{\boldsymbol{\alpha},\mathrm{ref}}\big(y|x\big)}{\widehat{Z}(x)} \exp\Big(\gamma r_{\theta^\star}(x, y)\Big), \tag{14}$$

*where*

$$\widehat{\pi}_{\boldsymbol{\alpha},\mathrm{ref}}(y|x) = \frac{\prod_{i=1}^{K} \pi_{\mathrm{ref},i}^{\alpha_i}(y|x)}{F_{\boldsymbol{\alpha}}(x)}, \quad F_{\boldsymbol{\alpha}}(x) = \sum_{y \in \mathcal{Y}} \prod_{i=1}^{K} \pi_{\mathrm{ref},i}^{\alpha_i}(y|x), \tag{15}$$

*and $\widehat{Z}(x) = \sum_y \widehat{\pi}_{\boldsymbol{\alpha},\mathrm{ref}}(y|x) \exp\Big(\gamma r_{\theta^\star}(x, y)\Big)$. The maximum objective value is $\frac{1}{\gamma} \log \Big( \sum_y \prod_{i=1}^{K} \pi_{\mathrm{ref},i}^{\alpha_i}(y|x) \exp\big(\gamma r(x, y)\big)\Big)$.*

Note that this result does not rely on the assumptions stated in Subsection 4.2 and in fact holds in greater generality. Using Theorem 5.1, we can consider the following optimization problem for the multiple reference models scenario.

$$\mathbb{E}_{Y \sim \pi(\cdot|x)}\big[r_{\theta^\star}(x, Y)\big] - \frac{1}{\gamma}\mathrm{KL}\big(\pi(\cdot|x)\|\widehat{\pi}_{\boldsymbol{\alpha},\mathrm{ref}}(\cdot|x)\big), \tag{16}$$

where $\widehat{\pi}_{\boldsymbol{\alpha},\mathrm{ref}}(y|x)$ is defined in (15) as generalized escort reference policy. The algorithm of reverse KL-regularized RLHF with two reference models is presented in App. A.

## 5.2 Main Results for RLHF via RKL

In this section, we provide our main theoretical results for the RLHF algorithm with multiple reference models based on reverse KL-regularization. Using the convexity of reverse KL divergence, we can provide an upper bound on the sub-optimality gap. Furthermore, we assume that Assumption 4.4 holds under $\widehat{\pi}_{\boldsymbol{\alpha},\mathrm{ref}}(\cdot|x)$ as reference policy with $C_{\boldsymbol{\alpha},\varepsilon_{\mathrm{rkl}}}$. First, we can derive the following upper bound on the sub-optimality gap of the RLHF algorithm with multiple reference models.

**Theorem 5.2.** *Under Assumption 4.1, 4.2 and 4.4, the following upper bound holds on the sub-optimality gap with probability at least $(1 - \delta)$ for $\delta \in (0, 1/2)$,*

$$\mathcal{J}^{\gamma}(\pi_{\theta^{\star}}^{\gamma}(\cdot|x), \pi_{\hat{\theta}}^{\gamma}(\cdot|x)) \leq \gamma C_{\boldsymbol{\alpha},\varepsilon_{\text{rkl}}} 128 e^{4R_{\max}} R_{\max}^2 \frac{\log(|\mathcal{R}|/\delta)}{n}. \tag{17}$$

Using Theorem 5.2, we can provide the upper bound on the optimal gap under the RLHF algorithm.

**Theorem 5.3.** *Under Assumption 4.1, 4.2 and 4.4, there exists constant $C > 0$ such that the following upper bound holds on optimality gap of reverse KL-regularized RLHF with probability at least $(1 - \delta)$ for $\delta \in (0, 1/2)$,*

$$\mathcal{J}(\pi_{\theta^{\star}}^{\gamma}(\cdot|x), \pi_{\hat{\theta}}^{\gamma}(\cdot|x)) \leq \gamma C_{\boldsymbol{\alpha},\varepsilon_{\text{rkl}}} 128 e^{4R_{\max}} R_{\max}^2 \frac{\log(|\mathcal{R}|/\delta)}{n} + C 8 R_{\max} e^{2R_{\max}} \sqrt{\frac{2 C_{\boldsymbol{\alpha},\varepsilon_{\text{rkl}}} \log(|\mathcal{R}|/\delta)}{n}}.$$

**Remark 5.4** (Sample Complexity). *We can observe sample complexity of $O(1/n)$ for the sub-optimality gap and $O(1/\sqrt{n})$ for the optimality gap from Theorem 5.2 and Theorem 5.3, respectively.*

# 6 RLHF from Multiple Reference Models via Forward KL Divergence

In this section, inspired by [Wang et al., 2024a], we extend the RLHF from multiple reference models based on reverse KL-regularization Le et al. [2024] to forward KL-regularization. Similar, to Section 5, we are focused on situations involving $K$ reference policies $\left\{\pi_{\text{ref},i}\right\}_{i=1}^{K}$ where the latent reward model among all reference policies is the same. All proof details are deferred to Appendix E.

## 6.1 Solution of RLHF under multiple reference models via FKL

Inspired by [Le et al., 2024, Wang et al., 2024a], our objective can be formulated as a multiple reference models RLHF objective,

$$\max_{\pi} \mathbb{E}_{Y \sim \pi(\cdot|x)}\left[r(x, Y)\right] - \frac{1}{\gamma}\left(\sum_{i=1}^{K} \beta_i \text{KL}\left(\pi_{\text{ref},i}(\cdot|x) \| \pi(\cdot|x)\right)\right), \tag{18}$$

where $\beta_i$ are weighting coefficients for each reference policy and $\sum_{i=1}^{K} \beta_i = 1$. This objective was explored in previous studies, leading to enhancements in pure RL problems Le et al. [2022]. However, addressing this optimization problem in LLMs through reward learning and RL finetuning poses similar challenges to (3). Our goal is to derive a closed-form solution for the multi-reference RLHF objective in (18).

We now provide the implicit solution of the RLHF with multiple references.

**Theorem 6.1.** *Consider the following objective function for RLHF with multiple reference models,*

$$\max_{\pi} \mathbb{E}_{Y \sim \pi(\cdot|x)}\left[r_{\theta^{\star}}(x, Y)\right] - \frac{1}{\gamma}\left(\sum_{i=1}^{K} \beta_i \text{KL}\left(\pi_{\text{ref},i}(\cdot|x) \| \pi(\cdot|x)\right)\right),$$

*where $\sum_{i=1}^{K} \beta_i = 1$ and $\beta_i \in (0, 1)$ for $i \in [K]$. Then, the implicit solution of the multiple reference models objective function for RLHF is,*

$$\tilde{\pi}_{\theta^{\star}}^{\gamma}(y|x) = \frac{\bar{\pi}_{\boldsymbol{\beta},\text{ref}}(y|x)}{\gamma\left(\tilde{Z}(x) - r_{\theta^{\star}}(x, y)\right)}, \tag{19}$$

*where $\bar{\pi}_{\boldsymbol{\beta},\text{ref}}(y|x) = \sum_{i=1}^{K} \beta_i \pi_{\text{ref},i}(y|x)$, and $\tilde{Z}(x)$ is the solution to $\int_{y \in \mathcal{Y}} \tilde{\pi}_{\theta^{\star}}^{\gamma}(y|x) = 1$ for a given $x \in \mathcal{X}$.*

Using Theorem 6.1, we can consider the following optimization problem for forward KL-regularized RLHF under multiple reference model scenario,

$$\mathbb{E}_{Y \sim \pi(\cdot|x)}\left[r_{\theta^{\star}}(x, Y)\right] - \frac{1}{\gamma}\text{KL}\left(\bar{\pi}_{\boldsymbol{\beta},\text{ref}}(\cdot|x) \| \pi(\cdot|x)\right), \tag{20}$$

where $\bar{\pi}_{\boldsymbol{\beta},\text{ref}}(y|x)$ is defined in Theorem 6.1 as weighted reference policy. The algorithm of forward KL-regularized RLHF with two reference models is presented in App. A.

Table 1: Comparison of Various Works in Theoretical Foundation of RLHF: Key features include support for RKL sub-optimality gap, RKL optimality gap, FKL sub-optimality gap, and FKL optimality gap and their Sample Complexities for each scenario.

| Work | RKL Sub-optimality Gap (Sample Complexity) | RKL Optimality Gap (Sample Complexity) | FKL Sub-optimality Gap (Sample Complexity) | FKL Optimality Gap (Sample Complexity) |
|---|---|---|---|---|
| Song et al. [2024] | ✓ $O(1/\sqrt{n})$ | ✗ | ✗ | ✗ |
| Zhao et al. [2024] | ✓ $O(1/n)$ | ✗ | ✗ | ✗ |
| Chang et al. [2024] | ✗ | ✓ $O(1/\sqrt{n})$ | ✗ | ✗ |
| Xiong et al. [2024] | ✓ $O(1/\sqrt{n})$ | ✗ | ✗ | ✗ |
| **Our Work** | ✓ $O(1/n)$ | ✓ $O(1/\sqrt{n})$ | ✓ $O(1/\sqrt{n})$ | ✓ $O(1/\sqrt{n})$ |

## 6.2 Main Results for RLHF with FKL

This section presents our core theoretical analysis of forward KL-regularized RLHF under the multiple reference model setting. We begin by leveraging KL divergence's convex properties to establish an upper bound on the sub-optimality gap. Throughout this section, we consider $\tilde{\pi}_{\hat{\theta}}^{\gamma}(y|x) = \frac{\bar{\pi}_{\boldsymbol{\beta},\mathrm{ref}}(y|x)}{\gamma(\bar{Z}_{\hat{\theta}}(x) - r_{\hat{\theta}}(x,y))}$ and $\tilde{\pi}_{\theta^{\star}}^{\gamma}(y|x) = \frac{\bar{\pi}_{\boldsymbol{\beta},\mathrm{ref}}(y|x)}{\gamma(\bar{Z}_{\theta^{\star}}(x) - r_{\theta^{\star}}(x,y))}$. Furthermore, we assume that Assumption 4.5 holds under $\bar{\pi}_{\boldsymbol{\beta},\mathrm{ref}}(y|x)$ as reference policy with $C_{\boldsymbol{\beta},\varepsilon_{\mathrm{rkl}}}$. First, we derive an upper bound for the sub-optimality gap in the multiple reference forward KL-regularized RLHF setting.

**Theorem 6.2.** *Under Assumption 4.1, 4.2 and 4.4, the following upper bound holds on the sub-optimality gap with probability at least $(1 - \delta)$ for $\delta \in (0, 1)$,*

$$\tilde{\mathcal{J}}^{\gamma}(\tilde{\pi}_{\theta^{\star}}^{\gamma}(\cdot|x), \tilde{\pi}_{\hat{\theta}}^{\gamma}(\cdot|x)) \leq 16 C_{\boldsymbol{\beta},\varepsilon_{\mathrm{fkl}}} e^{2R_{\max}} R_{\max} \sqrt{\frac{\log(|\mathcal{R}|/\delta)}{n}}. \tag{21}$$

Using Theorem 6.2, we can provide the upper bound on the optimal gap under the multiple reference forward KL-regularized RLHF setting.

**Theorem 6.3.** *Under Assumption 4.1, 4.2 and 4.4, the following upper bound holds on optimality gap of the multiple reference forward KL-regularized RLHF algorithm with probability at least $(1 - \delta)$ for $\delta \in (0, 1)$,*

$$\tilde{\mathcal{J}}(\tilde{\pi}_{\theta^{\star}}^{\gamma}(\cdot|x), \tilde{\pi}_{\hat{\theta}}^{\gamma}(\cdot|x)) \leq 16 C_{\boldsymbol{\beta},\varepsilon_{\mathrm{fkl}}} e^{2R_{\max}} R_{\max} \sqrt{\frac{\log(|\mathcal{R}|/\delta)}{n}} + \frac{\max\left(|\log(C_{\boldsymbol{\beta},\varepsilon_{\mathrm{fkl}}})|, \log(\gamma R_{\max} + 1)\right)}{\gamma}.$$

**Remark 6.4** (Sample Complexity). *Choosing $\gamma = n$, we have sample complexity $O(1/\sqrt{n})$ on optimality gap from Theorem 6.3. We can also observe the sample complexity of $O(1/\sqrt{n})$ for the sub-optimality gap.*

## 7 Discussion

In this section we provide theoretical comparison with single reference model in terms of sample complexity, $R_{\max}$ and coverage constant. We also extend our framework to DPO. Further discussion, e.g., coverage assumption and comparison of RKL with FKL, are provided in App. G.

**Theoretical Comparison with Single-Reference Models:** Our theoretical results extend to the single-reference model setting, enabling comparison with existing work in this domain. The RKL-regularized RLHF framework and its sub-optimality gap have been investigated by Song et al. [2024] and Zhao et al. [2024], who established sample complexity bounds. Song et al. [2024] derived a sub-optimality gap sample complexity of $O(1/\sqrt{n})$, which Zhao et al. [2024] later improved to $O(1/n)$, demonstrating the effectiveness of RKL regularization. Note that, in [Zhao et al., 2024], it is shown that when the error tolerance $\epsilon$ is sufficiently small, the sample complexity follows an $O(1/\epsilon)$ relationship. This corresponds to $O(1/n)$, where $n$ represents the dataset size. In comparison with [Zhao et al., 2024], we proposed an approach based on functional derivative and convexity of KL

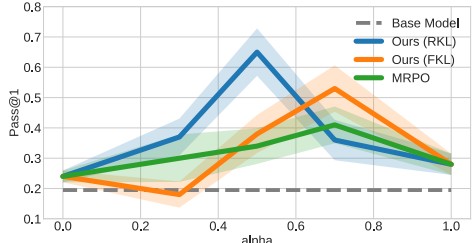 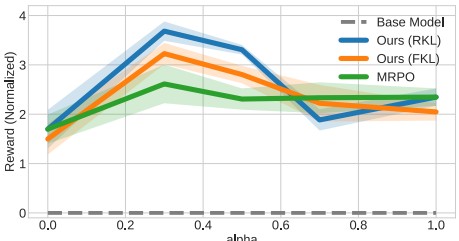

(a) Mean and 95% CI for pass@1 performance on GSM8K using policy gradient algorithms.

(b) Mean normalized reward and 95% CI on the UltraFeedback dataset, using offline RLHF.

Figure 1: In both online and offline RL, our analytical RKL objective outperforms both the MRPO approximation and single reference objective ($\alpha = 0$).

divergence. Our approach is more general and can be applied to the forward KL-regularized RLHF framework. There are also some works on similar algorithms to RLHF. Additionally, Chang et al. [2024] proposed an algorithm integrating offline and online preference datasets in RLHF, analyzing its optimality gap sample complexity under RKL regularization. The general reverse KL-regularized RLHF framework under general preference models is studied by Xiong et al. [2024] and a sample complexity of $O(1/\sqrt{n})$ for sub-optimality gap is derived. To the best of our knowledge, the sample complexity of the optimality gap and sub-optimality gap for forward KL-regularization have not been studied in the literature. Furthermore, in [Huang et al., 2024], KL-divergence and $\chi^2$-divergence are considered as regularizers, and the sample complexity on optimality gap for $\chi^2$-DPO are studied. We summarized our comparison with different works related to the theoretical study of RLHF in Table 1.

**Comparison in terms of $R_{\max}$ and coverage constant:** In Table 1, we compared different methods in terms of their sample complexity bounds. Regarding the dependency on $R_{\max}$, we observe that all existing bounds for RLHF with RKL regularization scale as $O(\exp(R_{\max}))$ [Song et al., 2024, Zhao et al., 2024, Chang et al., 2024, Xiong et al., 2024]. This exponential dependency arises directly from Lemma B.1, reflecting the inherent non-linearity introduced by the sigmoid function in the Bradley–Terry model. Additionally, concerning the coverage constant, the upper bounds under RKL regularization scale as $O(C_{\boldsymbol{\alpha}, \varepsilon_{\mathrm{rkl}}})$, highlighting the significant impact of coverage parameters on optimal and suboptimal regret bounds.

**Extension to DPO:** Our current results for reverse KL-regularized RLHF and forward KL-regularized RLHF can be extended to the DPO framework [Rafailov et al., 2023]. In particular, we can derive the following DPO function for reverse KL-regularized under multiple reference models scenario using Theorem 5.1,

$$\pi_{\mathrm{DPO}, \widehat{\theta}}^{\mathrm{RKL}} = \arg\max_{\pi_\theta \in \Pi} \sum_{i=1}^{n} \log\left[\sigma\left(\frac{1}{\gamma}\log(\frac{\pi_\theta(y_i^w|x_i)}{\pi_{\boldsymbol{\alpha},\mathrm{ref}}(y_i^w|x_i)}) - \frac{1}{\gamma}\log(\frac{\pi_\theta(y_i^l|x_i)}{\pi_{\boldsymbol{\alpha},\mathrm{ref}}(y_i^l|x_i)})\right)\right]. \quad (22)$$

For forward KL-regularized DPO, we can combine Theorem 6.1 with the approach outlined in [Wang et al., 2024a], to derive the DPO function,

$$\pi_{\mathrm{DPO}, \widehat{\theta}}^{\mathrm{FKL}} = \arg\max_{\pi_\theta \in \Pi} \sum_{i=1}^{n} \log\left[\sigma\left(\frac{1}{\gamma}\frac{\pi_{\boldsymbol{\beta},\mathrm{ref}}(y_i^l|x_i)}{\pi_\theta(y_i^l|x_i)} - \frac{1}{\gamma}\frac{\pi_{\boldsymbol{\beta},\mathrm{ref}}(y_i^w|x_i)}{\pi_\theta(y_i^w|x_i)}\right)\right]. \quad (23)$$

Furthermore, we derive optimality gap under bounded implicit reward assumptions in App. F.

## 8 Experiments

To support our theoretical findings, we conducted two sets of experiments: one using an online policy gradient algorithm, and another using an offline RLHF algorithm. Together, these experiments are designed to cover the primary use cases of KL-constrained RL optimization in the LLM post-training setting. Our experiments address two goals:

1. Evaluating the benefits of using multiple reference models versus a single reference.
2. Comparing our exact analytical solution to the approximation proposed by Le et al. [2024].

**Online RL.** Since our theory applies to general KL-constrained RL - not only to settings with learned reward models, as in standard RLHF - we ran an experiment on the GSM8K dataset [Cobbe et al., 2021] using GRPO [Shao et al., 2024], a policy gradient method. This setup uses a solution verifier as the reward model, avoiding complications from learned rewards and letting us focus on the effect of multiple reference models during training. We trained the instruction-tuned 0.5B model from the Qwen 2.5 family [Yang et al., 2024], and used the 1.5B math-specialized model from the same family as a second reference. For each value of $\alpha \in \{0.0, 0.3, 0.5, 0.7, 1.0\}$, for FKL we consider $\beta = \alpha$, we trained models using the following regularization: (1) *Normalized geometric mean* as in our multi-reference RKL objective, (2) *Arithmetic mean* as an approximation of our multi-reference FKL objective, and (3) *MRPO approximation* [Le et al., 2024] of the multi-reference RKL objective.

**Offline RL.** This experiment compares our exact analytical solution to MRPO [Le et al., 2024] in an offline RLHF setting. We trained the instruction-tuned 0.5B Qwen 2.5 model using the UltraFeedback dataset [Cui et al., 2023], with the 1.5B Qwen 2.5 model as the second reference. This can be seen as a combination of knowledge distillation [Gu et al., 2023] and RLHF. Evaluation was performed using the Skywork-Reward-Llama-3.1-8B-v0.2 reward model [Liu et al., 2024]. Here again we compared three training algorithms: (1) *DPO* [Rafailov et al., 2023] using the normalized geometric mean of reference policies, (2) *DPO based on FKL divergence* as proposed by [Wang et al., 2024a] using the arithmetic mean, and (3) *MRPO* version of DPO [Le et al., 2024].

To validate that our algorithm works at a larger scale, we also applied it to the Qwen 2.5 7B model. We trained this model on the UltraFeedback Cui et al. [2023] dataset, and evaluated the trained model's win rate against the preferred answer using GPT-4o as LLM-as-a-Judge. We follow the standard protocol of first performing SFT on the dataset before the DPO step. As a second reference, we used Qwen 2.5 14B Instruct. Due to the increased compute demands, we only experimented with $\alpha = 0.5$.

Table 2: Win rate against the preferred answer from the Ultrafeedback dataset. Combining both references leads to a substantial gain in performance.

| Model | Win rate |
|---|---|
| Base model (Qwen 2.5 7B) | 8.6% |
| SFT model | 43.4% |
| DPO (single reference – SFT model) | 56.4% |
| DPO (single reference – 14B model) | 59.8% |
| **Ours (DPO with both references)** | **66.1%** |

For more details on the experimental setting and discussion on the computational aspects of using multi-reference, see Appendix H.

## 9 Conclusion and Future Works

This work develops theoretical foundations for two Reinforcement Learning from Human Feedback (RLHF) frameworks: reverse KL-regularized and forward KL-regularized RLHF. We derive solutions for both frameworks under multiple reference scenarios and establish their sample complexity bounds. Our analysis reveals that while both algorithms share identical sample complexity for the optimality gap, the reverse KL-regularized RLHF achieves superior sample complexity for the sub-optimality gap.

The main limitation of our work lies in the assumption of bounded reward functions where some solutions are proposed to solve this limitation in App C. Promising directions for future work include: (a) extending our analysis to multiple-reference, KL-regularized RLHF with unbounded rewards or sub-Gaussian reward functions; (b) following [Wang et al., 2024a], investigating multiple-reference RLHF regularized by general $f$-divergences; (c) following [Sharifnassab et al., 2024], extending the analysis to preference models beyond the Bradley–Terry (BT) model; (d) following [Xu et al., 2025], combining our approach with doubly robust preference-optimization algorithms to mitigate model misspecification; and (e) extending inference-time algorithms (e.g., [Mroueh, 2024, Beirami et al., 2024, Aminian et al., 2025]) to the multiple-reference setting using our approach.

## Acknowledgment

The authors would like to thank Ahmad Beirami for his valuable comments and insightful feedback on our work. Gholamali Aminian acknowledges the support of the UKRI Prosperity Partnership Scheme (FAIR) under EPSRC Grant EP/V056883/1 and the Alan Turing Institute. Amir R. Asadi is supported by Leverhulme Trust grant ECF-2023-189 and Isaac Newton Trust grant 23.08(b).

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

# A  Algorithms

The RLHF algorithm with two reference models is shown in Algorithm 1. Furthermore, the forward KL-regularized RLHF algorithm with two reference models is shown in Algorithm 2.

---

**Algorithm 1** Reverse KL-regularized RLHF with Two Reference Models

---

**Require:** $\gamma, \alpha, \pi_{\text{ref},1}, \pi_{\text{ref},2}\ \Theta$
1: **for** $i = 1, \ldots, m$ **do**
2:     Sample prompt $\tilde{x}_i \sim \rho$ and 2 responses with their preference $\tilde{y}_i^w, \tilde{y}_i^l \sim \widehat{\pi}_{\alpha,\text{ref}}(\cdot|x) \propto \pi_{\text{ref},1}^{\alpha}(\cdot|\tilde{x}_i)\pi_{\text{ref},2}^{1-\alpha}(\cdot|\tilde{x}_i)$.
3: **end for**
4: Compute the MLE estimator of the reward function based on $D_n = \{(\tilde{x}_i, \tilde{y}_i^w, \tilde{y}_i^l)\}_{i=1}^n$:

$$\widehat{\theta} \leftarrow \arg\max_{\theta} \mathcal{L}(\theta, D_n),$$

5: Compute the RLHF output based on (16): $\pi_{\widehat{\theta}}^{\gamma}(\cdot|\cdot) \propto \widehat{\pi}_{\alpha,\text{ref}}(\cdot|x)\exp(\gamma r_{\widehat{\theta}}(\cdot,\cdot))$.

---

**Algorithm 2** Forward KL-regularized RLHF with Two Reference Models

---

**Require:** $\gamma, \beta, \pi_{\text{ref},1}, \pi_{\text{ref},2}\ \Theta$
1: **for** $i = 1, \ldots, m$ **do**
2:     Sample prompt $\tilde{x}_i \sim \rho$ and 2 responses with their preference
    $\tilde{y}_i^w, \tilde{y}_i^l \sim \bar{\pi}_{\beta,\text{ref}}(\cdot|x) = \beta\pi_{\text{ref},1}(\cdot|\tilde{x}_i) + (1-\beta)\pi_{\text{ref},2}(\cdot|\tilde{x}_i)$.
3: **end for**
4: Compute the MLE estimator of the reward function based on $D_n = \{(\tilde{x}_i, \tilde{y}_i^w, \tilde{y}_i^l)\}_{i=1}^n$:

$$\widehat{\theta} \leftarrow \arg\max_{\theta} \mathcal{L}(\theta, D_n)$$

5: Compute the RLHF output based on (20).

---

# B  Technical Tools

In this section, we introduce the following technical tools and Lemmata.

**Lemma B.1** (Lemma C.2 from [Chang et al., 2024]). *Under Assumptions 4.1 and 4.2, we have with probability at least $1 - \delta$ that*

$$\mathbb{E}_{Y^l, Y^w \sim \pi_{\text{ref}}, \pi_{\text{ref}}} \left[\left(r_{\theta^\star}(x, Y^l) - r_{\theta^\star}(x, Y^w) - r_{\widehat{\theta}}(x, Y^l) + r_{\widehat{\theta}}(x, Y^w)\right)^2\right]$$
$$\leq \frac{128 R_{\max}^2 \exp(4R_{\max})\log(|\mathcal{R}|/\delta)}{n}. \tag{24}$$

**Lemma B.2** ([Boucheron et al., 2013]). *Assume that function $f(x) \in [0, B]$ is bounded. Then, we have,*

$$\mathbb{E}_{p(X)}[f(X)] - \mathbb{E}_{q(X)}[f(X)] \leq B\sqrt{\frac{\text{KL}(p(X)\|q(X))}{2}}. \tag{25}$$

**Lemma B.3.** *Assume that $\tilde{\pi}_r(y|x) \propto \pi_{\text{ref}}(y|x)\exp(\gamma r(x, y))$. Then, $\tilde{\pi}_{r+\Delta}(y|x) = \tilde{\pi}_r(y|x)$, where $\Delta$ is constant.*

*Proof.*

$$\tilde{\pi}_{r+\Delta}(y|x) = \frac{\pi_{\text{ref}}(y|x)\exp(\gamma(r(x, y) + \Delta))}{\mathbb{E}_{Y \sim \pi_{\text{ref}}(Y|x)}\big[\exp(\gamma(r(x, y) + \Delta))\big]}$$
$$= \frac{\pi_{\text{ref}}(y|x)\exp(\gamma r(x, y))}{\mathbb{E}_{Y \sim \pi_{\text{ref}}(Y|x)}\big[\exp(\gamma r(x, y))\big]}. \tag{26}$$

$\square$

# C  Assumption 4.1 and Assumption 4.2 Discussion

These Assumptions are common literature are common in the literature [Song et al., 2024, Zhan et al., 2023, Zhao et al., 2024, Chang et al., 2024, Xiong et al., 2024]. In particular, Assumption 4.1 is primarily to enable the use of concentration inequalities like Freedman's inequality [Boucheron et al., 2013], which require bounded differences (as in Lemma B.1). However, this assumption can be relaxed under certain growth conditions, as discussed in [Freedman, 1975]. Moreover, even when the original reward function is unbounded or sub-Gaussian—as is often the case in human preference modeling—it is possible to apply a monotonic, bounded transformation to the rewards. For instance, one can use the cumulative distribution function (CDF) of the reward under a reference model to normalize the rewards into a bounded range, as proposed in [Balashankar et al., 2025]. This approach also retains the essential ordering of preferences and supports handling sub-Gaussian behavior in the transformed space. Regarding finite class, we can apply covering number and relax this assumption as utilized in [Zhao et al., 2024]

# D  Proofs and Details of Section 5

**Lemma D.1.** *Let $\alpha_i \in [0,1]$ for all $i \in [k]$ and $\sum_{i=1}^{k} \alpha_i = 1$. For any distributions $Q_i$ for all $i \in [k]$ and $P$ over the space $\mathcal{X}$, such that $P \ll Q_i$, we have*

$$\sum_{i=1}^{k} \alpha_i \mathrm{KL}(P\|Q_i) = \mathrm{KL}\left(P\|(\{Q_i\}_{i=1}^{k})^{\boldsymbol{\alpha}}\right) - \log\left(\sum_{x\in\mathcal{A}} \Pi_{i=1}^{k} Q_i^{\alpha_i}(x)\right).$$

*Proof.* We have

$$\sum_{i=1}^{k} \alpha_i \mathrm{KL}(P\|Q_i) = \sum_{i=1}^{k} \alpha_i \left(\sum_{x\in\mathcal{A}} P(x)\log\left(\frac{P(x)}{Q_i(x)}\right)\right)$$

$$= \sum_{x\in\mathcal{A}} \sum_{i=1}^{k} P(x)\log\left(\frac{P^{\alpha_i}(x)}{Q_i^{\alpha_i}(x)}\right)$$

$$= \sum_{x\in\mathcal{A}} P(x)\log\left(\frac{P(x)}{\prod_{i=1}^{k} Q_i^{\alpha_i}(x)}\right)$$

$$= \mathrm{KL}\left(P\|(\{Q_i\}_{i=1}^{k})^{\boldsymbol{\alpha}}\right) - \log\left(\sum_{x\in\mathcal{A}} \Pi_{i=1}^{k} Q_i^{\alpha_i}(x)\right).$$

$\square$

**Lemma D.2.** *Let $\mathcal{A}$ be an arbitrary set and function $f : \mathcal{A} \to \mathbb{R}$ be such that*

$$\int_{x\in\mathcal{A}} \exp\left(-\frac{f(x)}{\lambda}\right) Q_X(x)\mathrm{d}x < \infty.$$

*Then for any $P_X$ defined on $\mathcal{A}$ such that $X \sim P_X$, we have*

$$\mathbb{E}[f(X)] + \lambda\mathrm{KL}(P_X\|Q_X) = \lambda\mathrm{KL}\left(P_X\|P_X^{\mathrm{Gibbs}}\right) - \lambda\log\left(\int_{x\in\mathcal{A}} \exp\left(-\frac{f(x)}{\lambda}\right) Q_X(x)\mathrm{d}x\right),$$

*where*

$$P_X^{\mathrm{Gibbs}}(x) := \frac{\exp\left(-\frac{f(x)}{\lambda}\right) Q_X(x)}{\int_{x\in\mathcal{A}} \exp\left(-\frac{f(x)}{\lambda}\right) Q_X(x)\mathrm{d}x}, \quad x \in \mathcal{A},$$

*is the Gibbs–Boltzmann distribution.*

*Proof.* We have

$$\mathbb{E}[f(X)] + \lambda\mathrm{KL}(P_X\|Q_X) = \int f(x)P_X(x)\mathrm{d}x + \lambda\int P_X(x)\log\left(\frac{P_X(x)}{Q_X(x)}\right)$$

$$= \lambda\int P_X(x)\log\left(\frac{P_X(x)}{\exp\left(-\frac{f(x)}{\lambda}\right)Q_X(x)}\right)$$

$$= \lambda\mathrm{KL}\left(P_X\|P_X^{\mathrm{Gibbs}}\right) - \lambda\log\left(\int_{x\in\mathcal{A}}\exp\left(-\frac{f(x)}{\lambda}\right)Q_X(x)\mathrm{d}x\right).$$

$\square$

**Theorem 5.1.** *Consider the following objective function for RLHF with multiple reference models,*

$$\max_\pi\left\{\mathop{\mathbb{E}}_{Y\sim\pi(\cdot|x)}\left[r_{\theta^\star}\left(x,Y\right)\right] - \frac{1}{\gamma}\left(\sum_{i=1}^K\alpha_i\mathrm{KL}\left(\pi(\cdot|x)\|\pi_{\mathrm{ref},i}(\cdot|x)\right)\right)\right\},$$

*where $\sum_{i=1}^K\alpha_i = 1$ and $\alpha_i\in(0,1)$ for $i\in[K]$. Then, the exact solution of the multiple reference models objective function for RLHF is,*

$$\pi_{\theta^\star}^\gamma(y|x) = \frac{\widehat{\pi}_{\boldsymbol{\alpha},\mathrm{ref}}\left(y|x\right)}{\widehat{Z}(x)}\exp\left(\gamma r_{\theta^\star}\left(x,y\right)\right),\tag{27}$$

*where*

$$\widehat{\pi}_{\boldsymbol{\alpha},\mathrm{ref}}(y|x) = \frac{\prod_{i=1}^K\pi_{\mathrm{ref},i}^{\alpha_i}(y|x)}{F_{\boldsymbol{\alpha}}(x)},$$

$$F_{\boldsymbol{\alpha}}(x) = \sum_{y\in\mathcal{Y}}\prod_{i=1}^K\pi_{\mathrm{ref},i}^{\alpha_i}(y|x),$$

*and*

$$\widehat{Z}(x) = \sum_y\widehat{\pi}_{\boldsymbol{\alpha},\mathrm{ref}}(y|x)\exp\left(\gamma r(x,y)\right).$$

*The maximum objective value is*

$$\frac{1}{\gamma}\log\left(\sum_y\prod_{i=1}^K\pi_{\mathrm{ref},i}^{\alpha_i}(y|x)\exp\left(\gamma r(x,y)\right)\right).$$

*Proof.* We can write

$$\mathbb{E}_{Y\sim\pi(\cdot|x)}\big[r_{\theta^\star}(x,Y)\big] - \frac{1}{\gamma}\Big(\sum_{i=1}^{K}\alpha_i\mathrm{KL}\big(\pi(\cdot|x)\|\pi_{\mathrm{ref},i}(\cdot|x)\big)\Big)$$

$$= \frac{1}{\gamma}\bigg(\gamma\mathbb{E}_{Y\sim\pi(\cdot|x)}\big[r_{\theta^\star}(x,Y)\big] - \Big(\sum_{i=1}^{K}\alpha_i\mathrm{KL}\big(\pi(\cdot|x)\|\pi_{\mathrm{ref},i}(\cdot|x)\big)\Big)\bigg) \tag{28}$$

$$= \frac{1}{\gamma}\bigg(\gamma\mathbb{E}_{Y\sim\pi(\cdot|x)}\big[r_{\theta^\star}(x,Y)\big] - \mathrm{KL}(\pi(\cdot|x)\|\widehat{\pi}_{\boldsymbol{\alpha},\mathrm{ref}}(y|x)) + \log F_{\boldsymbol{\alpha}}(x)\bigg) \tag{29}$$

$$= \frac{1}{\gamma}\Big(-\mathrm{KL}(\pi(\cdot|x)\|\pi_{\theta^\star}^{\gamma}(y|x)) + \log \widehat{Z}(x) + \log F_{\boldsymbol{\alpha}}(x)\Big) \tag{30}$$

$$= \frac{1}{\gamma}\bigg(-\mathrm{KL}(\pi(\cdot|x)\|\pi_{\theta^\star}^{\gamma}(y|x)) + \log\Big(\sum_y\prod_{i=1}^{K}\pi_{\mathrm{ref},i}^{\alpha_i}(y|x)\exp\big(\gamma r(x,y)\big)\Big)\bigg), \tag{31}$$

where (29) follows from Lemma D.1 and (30) follows from Lemma D.2. Clearly, the right side of (31) is maximized when the KL divergence is set to zero. Thus, the maximizing distribution $\pi(\cdot|x)$ is identical to $\pi_{\theta^\star}^{\gamma}(y|x)$, and the maximum objective value is $\frac{1}{\gamma}\log\Big(\sum_y\prod_{i=1}^{K}\pi_{\mathrm{ref},i}^{\alpha_i}(y|x)\exp\big(\gamma r(x,y)\big)\Big)$. $\square$

---

**Corollary D.3.** *Weighted multiple single reverse KL-regularized RLHF problem is an upper bound on multiple references reverse KL-regularized RLHF problem, i.e.,*

$$\max_{\pi}\left\{\mathbb{E}_{Y\sim\pi(\cdot|x)}\big[r_{\theta^\star}(x,Y)\big] - \frac{1}{\gamma}\Big(\sum_{i=1}^{K}\alpha_i\mathrm{KL}\big(\pi(\cdot|x)\|\pi_{\mathrm{ref},i}(\cdot|x)\big)\Big)\right\}$$

$$\leq \sum_{i=1}^{K}\alpha_i\max_{\pi}\left\{\mathbb{E}_{Y\sim\pi(\cdot|x)}\big[r_{\theta^\star}(x,Y)\big] - \frac{1}{\gamma}\Big(\mathrm{KL}\big(\pi(\cdot|x)\|\pi_{\mathrm{ref},i}(\cdot|x)\big)\Big)\right\}. \tag{32}$$

*Proof.* It can be shown that the maximum of objective function in Theorem 5.1 is,

$$\max_{\pi}\left\{\mathbb{E}_{Y\sim\pi(\cdot|x)}\big[r_{\theta^\star}(x,Y)\big] - \frac{1}{\gamma}\Big(\sum_{i=1}^{K}\alpha_i\mathrm{KL}\big(\pi(\cdot|x)\|\pi_{\mathrm{ref},i}(\cdot|x)\big)\Big)\right\}$$

$$= \frac{1}{\gamma}\log\Big(\mathbb{E}_{Y\sim\widehat{\pi}_{\boldsymbol{\alpha},\mathrm{ref}}(y|x)}\big[\exp\big(\gamma r_{\theta^\star}(x,Y)\big)\big]\Big) + \frac{1}{\gamma}\log F_{\boldsymbol{\alpha}}(x)$$

$$= \frac{1}{\gamma}\log\Big(\sum_y\prod_{i=1}^{K}\pi_{\mathrm{ref},i}^{\alpha_i}(y|x)\exp\big(\alpha_i\gamma r_{\theta^\star}(x,y)\big)\Big) \tag{33}$$

$$\leq \sum_{i=1}^{K}\frac{\alpha_i}{\gamma}\log\Big(\sum_y\pi_{\mathrm{ref},i}(y|x)\exp\big(\gamma r_{\theta^\star}(x,y)\big)\Big),$$

where the last inequality follows from Hölder's inequality. Note that,

$$\max_{\pi}\left\{\mathbb{E}_{Y\sim\pi(\cdot|x)}\big[r_{\theta^\star}(x,Y)\big] - \frac{1}{\gamma}\Big(\mathrm{KL}\big(\pi(\cdot|x)\|\pi_{\mathrm{ref},i}(\cdot|x)\big)\Big)\right\}$$

$$= \frac{1}{\gamma}\log\Big(\sum_y\pi_{\mathrm{ref},i}(y|x)\exp\big(\gamma r_{\theta^\star}(x,Y)\big)\Big). \tag{34}$$

Then, we have,

$$\max_{\pi} \left\{ \mathbb{E}_{Y \sim \pi(\cdot|x)} \left[ r_{\theta^\star}(x, Y) \right] - \frac{1}{\gamma} \Big( \sum_{i=1}^{K} \alpha_i \mathrm{KL}\big(\pi(\cdot|x) \| \pi_{\mathrm{ref},i}(\cdot|x)\big) \Big) \right\}$$

$$\leq \sum_{i=1}^{K} \alpha_i \max_{\pi} \left\{ \mathbb{E}_{Y \sim \pi(\cdot|x)} \left[ r_{\theta^\star}(x, Y) \right] - \frac{1}{\gamma} \Big( \mathrm{KL}\big(\pi(\cdot|x) \| \pi_{\mathrm{ref},i}(\cdot|x)\big) \Big). \right\} \tag{35}$$

Therefore, multiple single RLHF problem is an upper bound on multiple reference models RLHF problem. $\qquad\square$

**Remark D.4** (Choosing $\boldsymbol{\alpha}$). *The optimum $\boldsymbol{\alpha}$ for a given $x$, can be derived from the following optimization problem,*

$$\max_{\boldsymbol{\alpha}} \frac{1}{\gamma} \log \Big( \sum_{y} \prod_{i=1}^{K} \pi_{\mathrm{ref},i}^{\alpha_i}(y|x) \exp\big(\alpha_i \gamma r_{\theta^\star}(x, y)\big) \Big). \tag{36}$$

---

**Proposition D.5.** *For a given response, $x \in \mathcal{X}$, the following upper bound holds,*

$$\mathcal{J}^{\gamma}(\pi_{\theta^\star}^{\gamma}(\cdot|x), \pi_{\widehat{\theta}}^{\gamma}(\cdot|x)) \leq \int_{\mathcal{Y}} (r_{\theta^\star}(x, y) - r_{\widehat{\theta}}(x, y))(\pi_{\theta^\star}^{\gamma}(y|x) - \pi_{\widehat{\theta}}^{\gamma}(y|x))(\mathrm{d}y).$$

---

*Proof.* Note that $\mathrm{KL}(\pi(\cdot|x) \| \widehat{\pi}_{\boldsymbol{\alpha},\mathrm{ref}}(\cdot|x))$ is a convex function with respect to $\pi(\cdot|x)$. Therefore, $J_{\gamma}(\widehat{\pi}_{\boldsymbol{\alpha},\mathrm{ref}}(\cdot|x), \pi(\cdot|x))$ is a concave function with respect to $\pi(\cdot|x)$. First, we compute the functional derivative of $J_{\gamma}(\widehat{\pi}_{\boldsymbol{\alpha},\mathrm{ref}}(\cdot|x), \pi(\cdot|x))$ with respect to $\pi(\cdot|x)$,

$$\frac{\delta J_{\gamma}(\widehat{\pi}_{\boldsymbol{\alpha},\mathrm{ref}}(\cdot|x), \pi(\cdot|x))}{\delta \pi} = r_{\theta^\star}(x, y) - \frac{1}{\gamma} \log(\pi(\cdot|x)/\widehat{\pi}_{\boldsymbol{\alpha},\mathrm{ref}}(\cdot|x)) + \frac{1}{\gamma}. \tag{37}$$

Therefore, we have,

$$\mathcal{J}^{\gamma}(\pi_{\theta^\star}^{\gamma}(\cdot|x), \pi_{\widehat{\theta}}^{\gamma}(\cdot|x)) =$$

$$J_{\gamma}(\widehat{\pi}_{\boldsymbol{\alpha},\mathrm{ref}}(\cdot|x), \pi_{\theta^\star}^{\gamma}(\cdot|x)) - J_{\gamma}(\widehat{\pi}_{\boldsymbol{\alpha},\mathrm{ref}}(\cdot|x), \pi_{\widehat{\theta}}^{\gamma}(\cdot|x))$$

$$\leq \int_{\mathcal{Y}} \frac{\delta J_{\gamma}(\widehat{\pi}_{\boldsymbol{\alpha},\mathrm{ref}}(\cdot|x), \pi_{\widehat{\theta}}^{\gamma}(y|x))}{\delta \pi} (\pi_{\theta^\star}^{\gamma}(y|x) - \pi_{\widehat{\theta}}^{\gamma}(y|x))(\mathrm{d}y)$$

$$= \int_{\mathcal{Y}} \Big( r_{\theta^\star}(x, y) - \frac{1}{\gamma} \log(\pi_{\widehat{\theta}}^{\gamma}(y|x)/\widehat{\pi}_{\boldsymbol{\alpha},\mathrm{ref}}(\cdot|x)) + \frac{1}{\gamma} \Big)(\pi_{\theta^\star}^{\gamma}(y|x) - \pi_{\widehat{\theta}}^{\gamma}(y|x))(\mathrm{d}y) \tag{38}$$

$$= \int_{\mathcal{Y}} \Big( r_{\theta^\star}(x, y) - r_{\widehat{\theta}}(x, y) + \frac{1}{\gamma} \log(Z(x)) \Big)(\pi_{\theta^\star}^{\gamma}(y|x) - \pi_{\widehat{\theta}}^{\gamma}(y|x))(\mathrm{d}y)$$

$$= \int_{\mathcal{Y}} \Big( r_{\theta^\star}(x, y) - r_{\widehat{\theta}}(x, y) \Big)(\pi_{\theta^\star}^{\gamma}(y|x) - \pi_{\widehat{\theta}}^{\gamma}(y|x))(\mathrm{d}y).$$

It completes the proof. $\qquad\square$

---

**Lemma D.6.** *Consider the softmax policy, $\pi_r^{\gamma}(y|x) \propto \widehat{\pi}_{\boldsymbol{\alpha},\mathrm{ref}}(y|x) \exp(\gamma r(x, y))$. Then, the sensitivity of the policy with respect to reward function is,*

$$\frac{\partial \pi_r^{\gamma}}{\partial r}(r) = \gamma \pi_r^{\gamma}(y|x)(1 - \pi_r^{\gamma}(y|x)).$$

---

*Proof.* We have $\pi_r^{\gamma}(y|x) = \frac{\widehat{\pi}_{\boldsymbol{\alpha},\mathrm{ref}}(y|x) \exp(\gamma r(x,y))}{\mathbb{E}_{Y \sim \widehat{\pi}_{\boldsymbol{\alpha},\mathrm{ref}}(\cdot|x)}[\exp(\gamma r(x,Y))]}$. Using Chain rule, we have,

$$\frac{\partial \pi_r^{\gamma}}{\partial r}(r) = \gamma \frac{\widehat{\pi}_{\boldsymbol{\alpha},\mathrm{ref}}(y|x) \exp(\gamma r(x,y))}{\mathbb{E}_{Y \sim \widehat{\pi}_{\boldsymbol{\alpha},\mathrm{ref}}(\cdot|x)}[\exp(\gamma r(x,Y))]} - \frac{\gamma \widehat{\pi}_{\boldsymbol{\alpha},\mathrm{ref}}(y|x)^2 \exp(2\gamma r(x,y))}{\mathbb{E}_{Y \sim \widehat{\pi}_{\boldsymbol{\alpha},\mathrm{ref}}(\cdot|x)}[\exp(\gamma r(x,Y))]^2} \tag{39}$$

$$= \gamma \pi_r^{\gamma}(y|x)(1 - \pi_r^{\gamma}(y|x)).$$

$\square$

> **Theorem 5.2.** *Under Assumption 4.1, 4.2 and 4.4, the following upper bound holds on the sub-optimality gap with probability at least $(1-\delta)$ for $\delta \in (0, 1/2)$,*
>
> $$\mathcal{J}^{\gamma}(\pi_{\theta^\star}^{\gamma}(\cdot|x), \pi_{\widehat{\theta}}^{\gamma}(\cdot|x))$$
>
> $$\leq \gamma C_{\boldsymbol{\alpha}, \varepsilon_{\mathrm{rkl}}} 128 e^{4R_{\max}} R_{\max}^2 \frac{\log(|\mathcal{R}|/\delta)}{n}.$$

*Proof.* Using Proposition D.5, we have,

$$\mathcal{J}^{\gamma}(\pi_{\theta^\star}^{\gamma}(\cdot|x), \pi_{\widehat{\theta}}^{\gamma}(\cdot|x))$$
$$\leq \int_{\mathcal{Y}} (r_{\theta^\star}(x,y) - r_{\widehat{\theta}}(x,y))(\pi_{\theta^\star}^{\gamma}(y|x) - \pi_{\widehat{\theta}}^{\gamma}(y|x))(\mathrm{d}y). \tag{40}$$

Note that, as the integral in (40) is over $\mathcal{Y}$, therefore, we have,

$$\mathcal{J}^{\gamma}(\pi_{\theta^\star}^{\gamma}(\cdot|x), \pi_{\widehat{\theta}}^{\gamma}(\cdot|x))$$
$$\leq \int_{\mathcal{Y}} (r_{\theta^\star}(x,y) - r_{\widehat{\theta}}(x,y) - h(x))(\pi_{\theta^\star}^{\gamma}(y|x) - \pi_{\widehat{\theta}}^{\gamma}(y|x))(\mathrm{d}y), \tag{41}$$

where $h(x)$ is an arbitrary function over $\mathcal{X}$. Note that $\pi_{\theta^\star}^{\gamma}(y|x)$ and $\pi_{\widehat{\theta}}^{\gamma}(y|x)$ are function of $r_{\theta^\star}(x,y)$ and $r_{\widehat{\theta}}(x,y)$, respectively. Furthermore, softmax policies are shift invariant, Lemma B.3, i.e., $\pi_{\theta^\star}^{\gamma}(y|x) \propto \widehat{\pi}_{\boldsymbol{\alpha},\mathrm{ref}}(\cdot|x) \exp(\gamma(r_{\theta}^\star(x,y) - h(x)))$ where $h(x)$ is a function dependent on $x$. Therefore, we can apply the mean-value theorem to $(\pi_{\theta^\star}^{\gamma}(y|x) - \pi_{\widehat{\theta}}^{\gamma}(y|x))(\mathrm{d}y)$ with respect to reward function $r(x,y)$. Therefore, we have for a given $h(x)$,

$$(\pi_{\theta^\star}^{\gamma}(y|x) - \pi_{\widehat{\theta}}^{\gamma}(y|x)) = \frac{\partial \pi(\cdot|x)}{\partial r}(r_{\lambda})(r_{\theta^\star}(x,y) - r_{\widehat{\theta}}(x,y) - h(x))$$
$$= \gamma \pi_{r_{\lambda}}(\cdot|x)(1 - \pi_{r_{\lambda}}(\cdot|x)(r_{\theta^\star}(x,y) - r_{\widehat{\theta}}(x,y) - h(x)), \tag{42}$$

where $r_{\lambda} = \lambda(r_{\theta^\star}(x,y) - h(x)) + (1-\lambda)r_{\widehat{\theta}}(x,y)$ for some $\lambda \in [0,1]$ and $\pi_{r_{\lambda}}(\cdot|x) \propto \widehat{\pi}_{\boldsymbol{\alpha},\mathrm{ref}}(\cdot|x) \exp(\gamma r_{\lambda}(x,y))$. Applying (42) in (41), we have,

$$\mathcal{J}^{\gamma}(\pi_{\theta^\star}^{\gamma}(\cdot|x), \pi_{\widehat{\theta}}^{\gamma}(\cdot|x))$$
$$\leq \gamma \int_{\mathcal{Y}} (r_{\theta^\star}(x,y) - r_{\widehat{\theta}}(x,y))^2 \pi_{r_{\lambda}}(\cdot|x)(1 - \pi_{r_{\lambda}}(\cdot|x))(\mathrm{d}y)$$
$$\leq \gamma \int_{\mathcal{Y}} (r_{\theta^\star}(x,y) - r_{\widehat{\theta}}(x,y))^2 \pi_{r_{\lambda}}(\cdot|x)(\mathrm{d}y) \tag{43}$$
$$\leq C_{\boldsymbol{\alpha}, \varepsilon_{\mathrm{rkl}}} \gamma \int_{\mathcal{Y}} (r_{\theta^\star}(x,y) - r_{\widehat{\theta}}(x,y) - h(x))^2 \widehat{\pi}_{\boldsymbol{\alpha},\mathrm{ref}}(\cdot|x)(\mathrm{d}y).$$

Choosing $h(x) = \mathbb{E}_{Y^l \sim \widehat{\pi}_{\boldsymbol{\alpha},\mathrm{ref}}(\cdot|x)}[r_{\theta^\star}(x,Y^l) - r_{\widehat{\theta}}(x,Y^l)]$, applying Jensen inequality and Lemma B.1, we have,

$$\mathcal{J}^{\gamma}(\pi_{\theta^\star}^{\gamma}(\cdot|x), \pi_{\widehat{\theta}}^{\gamma}(\cdot|x))$$
$$\leq C_{\boldsymbol{\alpha}, \varepsilon_{\mathrm{rkl}}} \gamma \int_{\mathcal{Y}} (r_{\theta^\star}(x,y^w) - r_{\widehat{\theta}}(x,y^w) - r_{\theta^\star}(x,y^l) + r_{\widehat{\theta}}(x,y^l))^2 \widehat{\pi}_{\boldsymbol{\alpha},\mathrm{ref}}(\cdot|x)(\mathrm{d}y^l)\widehat{\pi}_{\boldsymbol{\alpha},\mathrm{ref}}(\cdot|x)(\mathrm{d}y^w)$$
$$\leq \gamma C_{\boldsymbol{\alpha}, \varepsilon_{\mathrm{rkl}}} 128 e^{4R_{\max}} R_{\max}^2 \frac{\log(|\mathcal{R}|/\delta)}{n}. \tag{44}$$

This completes the proof. $\square$

> **Theorem 5.3.** *Under Assumption 4.1, 4.2 and 4.4, there exists constant $C > 0$ such that the following upper bound holds on the optimality gap of the reverse KL-regularized RLHF with probability at least $(1-\delta)$ for $\delta \in (0, 1/2)$,*
>
> $$\mathcal{J}(\pi_{\theta^\star}^\gamma(\cdot|x), \pi_{\widehat{\theta}}^\gamma(\cdot|x)) \leq \gamma C_{\boldsymbol{\alpha}, \varepsilon_{\mathrm{rkl}}} 128 e^{4R_{\max}} R_{\max}^2 \frac{\log(|\mathcal{R}|/\delta)}{n}$$
>
> $$+ C8 R_{\max} e^{2R_{\max}} \sqrt{\frac{2C_{\boldsymbol{\alpha}, \varepsilon_{\mathrm{rkl}}} \log(|\mathcal{R}|/\delta)}{n}}.$$

*Proof.* We have the following decomposition of the optimality gap,

$$\mathcal{J}(\pi_{\theta^\star}^\gamma(\cdot|x), \pi_{\widehat{\theta}}^\gamma(\cdot|x)) = \mathcal{J}^\gamma(\pi_{\theta^\star}^\gamma(\cdot|x), \pi_{\widehat{\theta}}^\gamma(\cdot|x)) + \frac{\mathrm{KL}(\pi_{\theta^\star}^\gamma(\cdot|x)\|\widehat{\pi}_{\boldsymbol{\alpha}, \mathrm{ref}}(\cdot|x)) - \mathrm{KL}(\pi_{\widehat{\theta}}^\gamma(\cdot|x)\|\widehat{\pi}_{\boldsymbol{\alpha}, \mathrm{ref}}(\cdot|x))}{\gamma}.$$
(45)

Now, we provide an upper bound on the second term using Lemma D.6 and a similar approach for choosing $h(x)$ in the proof of Theorem 5.2, we have for some $\lambda \in [0, 1]$,

$$\mathrm{KL}(\pi_{\theta^\star}^\gamma(\cdot|x)\|\widehat{\pi}_{\boldsymbol{\alpha}, \mathrm{ref}}(\cdot|x)) - \mathrm{KL}(\pi_{\widehat{\theta}}^\gamma(\cdot|x)\|\widehat{\pi}_{\boldsymbol{\alpha}, \mathrm{ref}}(\cdot|x))$$

$$= \int_{\mathcal{Y}} \frac{\partial \pi}{\partial r}(r_\lambda) \Big( \log \big( \frac{\pi_{r_\lambda}(\cdot|x)}{\widehat{\pi}_{\boldsymbol{\alpha}, \mathrm{ref}}(\cdot|x)} \big) + 1 \Big) (r_{\theta^\star}(x, y) - r_{\widehat{\theta}}(x, y) - h(x))(\mathrm{d}y)$$

$$= \gamma \int_{\mathcal{Y}} \pi_{r_\lambda}(\cdot|x)(1 - \pi_{r_\lambda}(\cdot|x)) \Big( \log \big( \frac{\pi_{r_\lambda}(\cdot|x)}{\widehat{\pi}_{\boldsymbol{\alpha}, \mathrm{ref}}(\cdot|x)} \big) + 1 \Big) (r_{\theta^\star}(x, y) - r_{\widehat{\theta}}(x, y) - h(x))(\mathrm{d}y)$$
(46)

$$\leq \gamma \sqrt{\int_{\mathcal{Y}} (1 - \pi_{r_\lambda}(\cdot|x))^2 \Big( \log \big( \frac{\pi_{r_\lambda}(\cdot|x)}{\widehat{\pi}_{\boldsymbol{\alpha}, \mathrm{ref}}(\cdot|x)} \big) + 1 \Big)^2 (\mathrm{d}y)}$$

$$\times \sqrt{\int_{\mathcal{Y}} \pi_{r_\lambda}(\cdot|x)^2 (r_{\theta^\star}(x, y) - r_{\widehat{\theta}}(x, y) - h(x))^2 (\mathrm{d}y)},$$

where, in the last inequality, we applied the Cauchy–Schwarz inequality. Using the fact that $\pi_{r_\lambda} \propto \widehat{\pi}_{\boldsymbol{\alpha}, \mathrm{ref}}(\cdot|x) \exp(\gamma r_\lambda)$ and Lemma B.1, we have,

$$\mathrm{KL}(\pi_{\widehat{\theta}}^\gamma(\cdot|x)\|\widehat{\pi}_{\boldsymbol{\alpha}, \mathrm{ref}}(\cdot|x)) - \mathrm{KL}(\pi_{\theta^\star}^\gamma(\cdot|x)\|\widehat{\pi}_{\boldsymbol{\alpha}, \mathrm{ref}}(\cdot|x))$$

$$\leq \gamma 8 \Big( 2\gamma R_{\max} + 1 \Big) R_{\max} \exp(2R_{\max}) \sqrt{\frac{2C_{\boldsymbol{\alpha}, \varepsilon_{\mathrm{rkl}}} \log(|\mathcal{R}|/\delta)}{n}}.$$
(47)

The final result holds by applying the union bound. $\square$

In the following, we compare the RLHF objective function under the multiple reference model policy, $\widehat{\pi}_{\boldsymbol{\alpha}, \mathrm{ref}}(\cdot|x)$, with $i$-th reference model, $\pi_{\mathrm{ref}, i}(\cdot|x)$. For this purpose, we bound the difference between these two RLHF objective functions in different scenarios.

> **Proposition D.7.** *Under Assumption 4.1, the following upper bound holds,*
>
> $$\tilde{J}_\gamma(\pi_{\boldsymbol{\alpha}, \mathrm{ref}}, \pi_{\theta^\star}^\gamma) - \tilde{J}_\gamma(\pi_{\mathrm{ref}, i}, \pi_{\theta^\star, i}^\gamma) \leq \frac{\exp(\gamma R_{\max}) - 1}{\gamma \sqrt{2}} \sqrt{\mathrm{KL}(\pi_{\boldsymbol{\alpha}, \mathrm{ref}}(\cdot|x)\|\pi_{\mathrm{ref}, i}(\cdot|x))}.$$

*Proof.* Note that, for a policy $\pi_{\mathrm{ref}}$ we have,

$$\tilde{J}_\gamma(\pi_{\mathrm{ref}}, \pi_{\theta^\star}^\gamma) = \frac{1}{\gamma} \log \Big[ \mathbb{E}_{\pi_{\mathrm{ref}}}[\exp(\gamma r_{\theta^\star}(x, y))] \Big].$$
(48)

Therefore, using the functional derivative, we have,

$$
\begin{aligned}
&\tilde{J}_\gamma(\pi_{\boldsymbol{\alpha},\mathrm{ref}}, \pi_{\theta^\star}^\gamma) - \tilde{J}_\gamma(\pi_{\mathrm{ref},i}, \pi_{\theta^\star,i}^\gamma) \\
&= \frac{1}{\gamma} \log\left[\mathbb{E}_{\pi_{\boldsymbol{\alpha},\mathrm{ref}}}[\exp(\gamma r_{\theta^\star}(x,y))]\right] - \frac{1}{\gamma}\log\left[\mathbb{E}_{\pi_{\mathrm{ref},i}}[\exp(\gamma r_{\theta^\star}(x,y))]\right] \\
&= \frac{1}{\gamma}\int_0^1 \int_{\mathcal{Y}} \frac{\exp(\gamma r_{\theta^\star}(x,y))}{\mathbb{E}_{\pi_{\mathrm{ref},\lambda}}[\exp(\gamma r_{\theta^\star}(x,y))]}\big(\pi_{\boldsymbol{\alpha},\mathrm{ref}} - \pi_{\mathrm{ref},i}\big)(\mathrm{d}y)\mathrm{d}\lambda \\
&= \frac{1}{\gamma}\int_0^1 \frac{1}{\mathbb{E}_{\pi_{\mathrm{ref},\lambda}}[\exp(\gamma r_{\theta^\star}(x,y))]}\int_{\mathcal{Y}}\exp(\gamma r_{\theta^\star}(x,y))\big(\pi_{\boldsymbol{\alpha},\mathrm{ref}} - \pi_{\mathrm{ref},i}\big)(\mathrm{d}y)\mathrm{d}\lambda \\
&\le \frac{\exp(\gamma R_{\max})-1}{\gamma}\sqrt{\frac{\mathrm{KL}(\pi_{\boldsymbol{\alpha},\mathrm{ref}}(\cdot|x)\|\pi_{\mathrm{ref},i}(\cdot|x))}{2}},
\end{aligned}
\tag{49}
$$

where $\pi_{\mathrm{ref},\lambda} = \pi_{\mathrm{ref},i} + \lambda\big(\pi_{\boldsymbol{\alpha},\mathrm{ref}} - \pi_{\mathrm{ref},i}\big)$ and the last inequality holds due to Lemma B.2. $\square$

# E   Proofs and Details of Section 6

> **Lemma E.1.** *Let $\beta_i \in [0, 1]$ for all $i \in [k]$ and $\sum_{i=1}^{k} \beta_i = 1$. For any distributions $Q_i$ for all $i \in [k]$ and $R$ such that $Q_i \ll P$, we have*
>
> $$\sum_{i=1}^{k} \beta_i \mathrm{KL}(Q_i \| P) = H\Big( \sum_{i=1}^{k} \beta_i Q_i \Big) - \sum_{i=1}^{k} \beta_i H(Q_i) + \mathrm{KL}\Big( \sum_{i=1}^{k} \beta_i Q_i \| P \Big).$$

*Proof.* We have,

$$\sum_{i=1}^{k} \beta_i \mathrm{KL}(Q_i \| P) \tag{50}$$

$$= \sum_{i=1}^{k} \beta_i Q_i \log(Q_i) - \beta_i Q_i \log(P)$$

$$= -\sum_{i=1}^{k} \beta_i H(Q_i) + \Big( \sum_{i=1}^{k} \beta_i Q_i \Big) \log \Big( \sum_{i=1}^{k} \beta_i Q_i \Big) - \Big( \sum_{i=1}^{k} \beta_i Q_i \Big) \log \Big( \sum_{i=1}^{k} \beta_i Q_i \Big) - \Big( \sum_{i=1}^{k} \beta_i Q_i \Big) \log(P) \tag{51}$$

$$= H\Big( \sum_{i=1}^{k} \beta_i Q_i \Big) - \sum_{i=1}^{k} \beta_i H(Q_i) + \Big( \sum_{i=1}^{k} \beta_i Q_i \Big) \log\Big( \sum_{i=1}^{k} \beta_i Q_i \Big) - \Big( \sum_{i=1}^{k} \beta_i Q_i \Big) \log(P) \tag{52}$$

$$= H\Big( \sum_{i=1}^{k} \beta_i Q_i \Big) - \sum_{i=1}^{k} \beta_i H(Q_i) + \mathrm{KL}\Big( \sum_{i=1}^{k} \beta_i Q_i \| P \Big). \tag{53}$$

$\square$

> **Theorem 6.1.** *Consider the following objective function for RLHF with multiple reference models,*
>
> $$\max_{\pi} \mathbb{E}_{Y \sim \pi(\cdot|x)} \big[ r_{\theta^\star}(x, Y) \big] - \frac{1}{\gamma} \Big( \sum_{i=1}^{K} \beta_i \mathrm{KL}\big( \pi_{\mathrm{ref},i}(\cdot|x) \| \pi(\cdot|x) \big) \Big),$$
>
> *where $\sum_{i=1}^{K} \beta_i = 1$ and $\beta_i \in (0, 1)$ for $i \in [K]$. Then, the implicit solution of the multiple reference models objective function for RLHF is,*
>
> $$\tilde{\pi}_{\theta^\star}^{\gamma}(y|x) = \frac{\bar{\pi}_{\boldsymbol{\beta},\mathrm{ref}}\big( y|x \big)}{\gamma \big( \tilde{Z}_{\theta^\star}(x) - r_{\theta^\star}(x, y) \big)},$$
>
> *where*
>
> $$\bar{\pi}_{\boldsymbol{\beta},\mathrm{ref}}(y|x) = \sum_{i=1}^{K} \beta_i \pi_{\mathrm{ref},i}(y|x),$$
>
> *and $\tilde{Z}_{\theta^\star}(x)$ is the solution to $\int_{y \in \mathcal{Y}} \tilde{\pi}_{\theta^\star}^{\gamma}(y|x) = 1$ for a given $x \in \mathcal{X}$.*

*Proof.* Using Lemma E.1, the objective function of forward KL-regularization under multiple reference model can be represented as,

$$\max_{\pi} \mathbb{E}_{Y \sim \pi(\cdot|x)} \big[ r_{\theta^\star}(x, Y) \big] - \frac{1}{\gamma} \mathrm{KL}\big( \bar{\pi}_{\boldsymbol{\beta},\mathrm{ref}}(\cdot|x) \| \pi(\cdot|x) \big),$$

where $\bar{\pi}_{\boldsymbol{\beta},\mathrm{ref}}(y|x) = \sum_{i=1}^{K} \beta_i \pi_{\mathrm{ref},i}(y|x)$. As the function is a concave function with respect to $\pi(\cdot|x)$, we can compute the derivative with respect to $\pi(\cdot|x)$. Therefore, using the functional derivative

under the constraint that $\pi(\cdot|x)$ is a probability measure with Lagrange multiplier, $\tilde{Z}_{\theta^\star}(x)$, we have at optimal solution that,

$$r_{\theta^\star}(x, y) + \frac{1}{\gamma} \frac{\bar{\pi}_{\boldsymbol{\beta},\text{ref}}(y|x)}{\tilde{\pi}_{\theta^\star}^\gamma(y|x)} - \tilde{Z}_{\theta^\star}(x) = 0. \tag{54}$$

Solving (54) results in the final solution, $\tilde{\pi}_{\theta^\star}^\gamma(y|x)$. $\qquad\square$

**Corollary E.2.** *Weighted multiple single forward KL-regularized RLHF problem is an upper bound on multiple references forward KL-regularized RLHF problem, i.e.,*

$$
\begin{aligned}
\max_\pi &\left\{ \mathop{\mathbb{E}}_{Y\sim\pi(\cdot|x)} \big[ r_{\theta^\star}(x, Y) \big] - \frac{1}{\gamma} \Big( \sum_{i=1}^K \beta_i \text{KL}\big(\pi_{\text{ref},i}(\cdot|x)\|\pi(\cdot|x)\big) \Big) \right\} \\
&\leq \sum_{i=1}^K \beta_i \max_\pi \left\{ \mathop{\mathbb{E}}_{Y\sim\pi(\cdot|x)} \big[ r_{\theta^\star}(x, Y) \big] - \frac{1}{\gamma} \Big( \text{KL}\big(\pi_{\text{ref},i}(\cdot|x)\|\pi(\cdot|x)\big) \Big) \right\}.
\end{aligned}
\tag{55}
$$

*Proof.* It holds due to maximum function property. $\qquad\square$

Assuming,

$$\tilde{\pi}_{\theta^\star}^\gamma(y|x) = \frac{\bar{\pi}_{\boldsymbol{\beta},\text{ref}}(y|x)}{\gamma\big(\tilde{Z}_{\theta^\star}(x) - r_{\theta^\star}(x, y)\big)},$$

we can provide the following property of $\tilde{Z}_{\theta^\star}(x)$, inspired by [Cohen, 2017].

---

**Lemma E.3.** *The following property holds for $\tilde{Z}_{\theta^\star}(x)$,*

- *For any $x \in \mathcal{X}$ where $\rho(x) > 0$, we have $\sup_{y\in\mathcal{Y}} r_{\theta^\star}(x, y) \leq \tilde{Z}_{\theta^\star}(x)$.*
- *Under Assumption 4.1, we have $\sup_{x\in\mathcal{X}} \tilde{Z}_{\theta^\star}(x) \leq R_{\max} + \frac{1}{\gamma}$.*

---

*Proof.* Using the following representation,

$$\tilde{\pi}_{\theta^\star}^\gamma(y|x) = \frac{\bar{\pi}_{\boldsymbol{\beta},\text{ref}}(y|x)}{\gamma\big(\tilde{Z}_{\theta^\star}(x) - r_{\theta^\star}(x, y)\big)},$$

we can conclude that for a given $x \in \mathcal{X}$, $\sup_{y\in\mathcal{Y}} r_{\theta^\star}(x, y) \leq \tilde{Z}_{\theta^\star}(x)$. Otherwise, $\tilde{\pi}_{\theta^\star}^\gamma(y|x)$ will be negative.

For the second part, let's proceed by contradiction. Suppose there exists some $x \in \mathcal{X}$ such that: $\tilde{Z}\theta^\star(x) > \sup y \in \mathcal{Y} r_{\theta^\star}(x, y) + \frac{1}{\gamma}$

Under this assumption, we can show that:

$$\int_y \tilde{\pi}_{\theta^\star}^\gamma(y|x)(\mathrm{d}y) < 1.$$

This contradicts the fundamental requirement that

$$\tilde{\pi}_{\theta^\star}^\gamma(y|x),$$

must be a probability distribution. Therefore, our initial assumption must be false. Consequently, for all $x \in \mathcal{X}$, we must have:

$$\tilde{Z}\theta^\star(x) \leq \sup_{y\in\mathcal{Y}} r_{\theta^\star}(x, y) + \frac{1}{\gamma}.$$

Taking the supremum of both sides with respect to $x$ completes the proof. $\qquad\square$

> **Proposition E.4.** *For a given response, $x \in \mathcal{X}$, the following upper bound holds,*
>
> $$\widetilde{\mathcal{J}}^\gamma(\tilde{\pi}_{\theta^\star}^\gamma(\cdot|x), \tilde{\pi}_{\hat{\theta}}^\gamma(\cdot|x)) \leq$$
>
> $$\int_{\mathcal{Y}} (r_{\theta^\star}(x,y) - r_{\hat{\theta}}(x,y))(\tilde{\pi}_{\theta^\star}^\gamma(y|x) - \tilde{\pi}_{\hat{\theta}}^\gamma(y|x))(\mathrm{d}y).$$

*Proof.* The proof is similar to Proposition D.5. Note that $\mathrm{KL}(\bar{\pi}_{\boldsymbol{\beta},\mathrm{ref}}(\cdot|x)\|\pi(\cdot|x))$ is a convex function with respect $\pi(\cdot|x)$. Therefore, $\tilde{J}_\gamma(\bar{\pi}_{\boldsymbol{\beta},\mathrm{ref}}(\cdot|x), \pi(\cdot|x))$ is a concave function with respect to $\pi(\cdot|x)$. First, we compute the functional derivative of $\tilde{J}_\gamma(\bar{\pi}_{\boldsymbol{\beta},\mathrm{ref}}(\cdot|x), \pi(\cdot|x))$ with respect to $\pi(\cdot|x)$,

$$\frac{\delta \tilde{J}_\gamma(\bar{\pi}_{\boldsymbol{\beta},\mathrm{ref}}(\cdot|x), \pi(\cdot|x))}{\delta\pi} = r_{\theta^\star}(x,y) + \frac{1}{\gamma}\frac{\bar{\pi}_{\boldsymbol{\beta},\mathrm{ref}}(\cdot|x)}{\pi(\cdot|x)}. \tag{56}$$

Therefore, we have,

$$\widetilde{\mathcal{J}}^\gamma(\tilde{\pi}_{\theta^\star}^\gamma(\cdot|x), \tilde{\pi}_{\hat{\theta}}^\gamma(\cdot|x)) \leq \int_{\mathcal{Y}} \left(r_{\theta^\star}(x,y) + \frac{1}{\gamma}\frac{\bar{\pi}_{\boldsymbol{\beta},\mathrm{ref}}(y|x)}{\tilde{\pi}_{\hat{\theta}}^\gamma(y|x)}\right)(\tilde{\pi}_{\theta^\star}^\gamma(y|x) - \tilde{\pi}_{\hat{\theta}}^\gamma(y|x))(\mathrm{d}y), \tag{57}$$

Using the fact that $\tilde{\pi}_{\hat{\theta}}^\gamma(y|x) = \frac{\bar{\pi}_{\boldsymbol{\beta},\mathrm{ref}}(y|x)}{\gamma(\tilde{Z}(x) - r_{\hat{\theta}}(x,y))}$,

$$\widetilde{\mathcal{J}}^\gamma(\tilde{\pi}_{\theta^\star}^\gamma(\cdot|x), \tilde{\pi}_{\hat{\theta}}^\gamma(\cdot|x)) \leq \int_{\mathcal{Y}} \left(r_{\theta^\star}(x,y) - r_{\hat{\theta}}(x,y) + \tilde{Z}(x)\right)(\tilde{\pi}_{\theta^\star}^\gamma(y|x) - \tilde{\pi}_{\hat{\theta}}^\gamma(y|x))(\mathrm{d}y)$$

$$= \int_{\mathcal{Y}} \left(r_{\theta^\star}(x,y) - r_{\hat{\theta}}(x,y)\right)(\tilde{\pi}_{\theta^\star}^\gamma(y|x) - \tilde{\pi}_{\hat{\theta}}^\gamma(y|x))(\mathrm{d}y), \tag{58}$$

where the last equality follows from the fact that $\tilde{Z}(x)$ is just dependent on $x$. $\qquad\square$

> **Theorem 6.2.** *Under Assumption 4.1, 4.2 and 4.4, the following upper bound holds on the sub-optimality gap with probability at least $(1-\delta)$ for $\delta \in (0,1)$,*
>
> $$\widetilde{\mathcal{J}}^\gamma(\tilde{\pi}_{\theta^\star}^\gamma(\cdot|x), \tilde{\pi}_{\hat{\theta}}^\gamma(\cdot|x)) \leq 16C_{\boldsymbol{\beta},\varepsilon_{\mathrm{fkl}}}e^{2R_{\max}}R_{\max}\sqrt{\frac{\log(|\mathcal{R}|/\delta)}{n}}.$$

*Proof.* From Proposition E.4, we have,

$$\widetilde{\mathcal{J}}^\gamma(\tilde{\pi}_{\theta^\star}^\gamma(\cdot|x), \tilde{\pi}_{\hat{\theta}}^\gamma(\cdot|x)) \leq \int_{\mathcal{Y}} \left(r_{\theta^\star}(x,y) - r_{\hat{\theta}}(x,y)\right)(\tilde{\pi}_{\theta^\star}^\gamma(y|x) - \tilde{\pi}_{\hat{\theta}}^\gamma(y|x))(\mathrm{d}y)$$

$$= \int_{\mathcal{Y}} \left(r_{\theta^\star}(x,y) - r_{\hat{\theta}}(x,y) - h(x)\right)(\tilde{\pi}_{\theta^\star}^\gamma(y|x) - \tilde{\pi}_{\hat{\theta}}^\gamma(y|x))(\mathrm{d}y)$$

$$= \int_{\mathcal{Y}} \left(r_{\theta^\star}(x,y) - r_{\hat{\theta}}(x,y) - h(x)\right)\bar{\pi}_{\boldsymbol{\beta},\mathrm{ref}}(y|x)\frac{(\tilde{\pi}_{\theta^\star}^\gamma(y|x) - \tilde{\pi}_{\hat{\theta}}^\gamma(y|x))}{\bar{\pi}_{\boldsymbol{\beta},\mathrm{ref}}(y|x)}(\mathrm{d}y)$$

$$\leq \sqrt{\int_{\mathcal{Y}} \left(r_{\theta^\star}(x,y) - r_{\hat{\theta}}(x,y) - h(x)\right)^2(\bar{\pi}_{\boldsymbol{\beta},\mathrm{ref}}(y|x))^2(\mathrm{d}y)}\sqrt{\int_{\mathcal{Y}} \frac{(\tilde{\pi}_{\theta^\star}^\gamma(y|x) - \tilde{\pi}_{\hat{\theta}}^\gamma(y|x))^2}{(\bar{\pi}_{\boldsymbol{\beta},\mathrm{ref}}(y|x))^2}(\mathrm{d}y)}$$

$$\leq \sqrt{\int_{\mathcal{Y}} \left(r_{\theta^\star}(x,y) - r_{\hat{\theta}}(x,y) - h(x)\right)^2\bar{\pi}_{\boldsymbol{\beta},\mathrm{ref}}(y|x)(\mathrm{d}y)}\sqrt{\int_{\mathcal{Y}} \frac{(\tilde{\pi}_{\theta^\star}^\gamma(y|x) - \tilde{\pi}_{\hat{\theta}}^\gamma(y|x))^2}{(\bar{\pi}_{\boldsymbol{\beta},\mathrm{ref}}(y|x))^2}(\mathrm{d}y)}$$

$$\leq 16C_{\boldsymbol{\beta},\varepsilon_{\mathrm{fkl}}}e^{2R_{\max}}R_{\max}\sqrt{\frac{\log(|\mathcal{R}|/\delta)}{n}}, \tag{59}$$

where the first, second, and last inequalities follow from the Cauchy–Schwarz inequality, $(\bar{\pi}_{\boldsymbol{\beta},\mathrm{ref}}(y|x))^2 \leq \bar{\pi}_{\boldsymbol{\beta},\mathrm{ref}}(y|x)$ and using Assumption 4.5 and Lemma B.1, respectively. $\qquad\square$

> **Theorem 6.3.** *Under Assumption 4.1, 4.2 and 4.4, there exists constant $D > 0$ such that the following upper bound holds on optimality gap of the multiple reference forward KL-regularized RLHF algorithm with probability at least $(1 - \delta)$ for $\delta \in (0, 1)$,*
>
> $$\tilde{\mathcal{J}}(\tilde{\pi}_{\theta^\star}^\gamma(\cdot|x), \tilde{\pi}_{\hat{\theta}}^\gamma(\cdot|x))$$
>
> $$\leq 16 C_{\boldsymbol{\beta}, \varepsilon_{\mathrm{fkl}}} e^{2R_{\max}} R_{\max} \sqrt{\frac{\log(|\mathcal{R}|/\delta)}{n}} + \frac{\max\left(|\log(C_{\boldsymbol{\beta}, \varepsilon_{\mathrm{fkl}}})|, \log(\gamma R_{\max} + 1)\right)}{\gamma}$$

*Proof.* We have the following decomposition of the optimality gap,

$$\tilde{\mathcal{J}}(\tilde{\pi}_{\theta^\star}^\gamma(\cdot|x), \tilde{\pi}_{\hat{\theta}}^\gamma(\cdot|x)) = \tilde{\mathcal{J}}^\gamma(\tilde{\pi}_{\theta^\star}^\gamma(\cdot|x), \tilde{\pi}_{\hat{\theta}}^\gamma(\cdot|x))$$
$$+ \frac{\mathrm{KL}(\bar{\pi}_{\boldsymbol{\beta}, \mathrm{ref}}(\cdot|x) \| \tilde{\pi}_{\hat{\theta}}^\gamma(\cdot|x)) - \mathrm{KL}(\bar{\pi}_{\boldsymbol{\beta}, \mathrm{ref}}(\cdot|x) \| \tilde{\pi}_{\theta^\star}^\gamma(\cdot|x))}{\gamma}. \tag{60}$$

For second term, using the fact that, $\tilde{\pi}_{\hat{\theta}}^\gamma(y|x) = \frac{\bar{\pi}_{\boldsymbol{\beta}, \mathrm{ref}}(y|x)}{\gamma(\tilde{Z}_{\hat{\theta}}(x) - r_{\hat{\theta}}(x, y))}$ and $\tilde{\pi}_{\theta^\star}^\gamma(y|x) = \frac{\bar{\pi}_{\boldsymbol{\beta}, \mathrm{ref}}(y|x)}{\gamma(\tilde{Z}_{\theta^\star}(x) - r_{\theta^\star}(x, y))}$, we have,

$$\frac{\mathrm{KL}(\bar{\pi}_{\boldsymbol{\beta}, \mathrm{ref}}(\cdot|x) \| \tilde{\pi}_{\hat{\theta}}^\gamma(\cdot|x)) - \mathrm{KL}(\bar{\pi}_{\boldsymbol{\beta}, \mathrm{ref}}(\cdot|x) \| \tilde{\pi}_{\theta^\star}^\gamma(\cdot|x))}{\gamma}$$

$$= \frac{\mathbb{E}_{Y \sim \bar{\pi}_{\boldsymbol{\beta}, \mathrm{ref}}(\cdot|x)}[\log(\gamma(\tilde{Z}_{\hat{\theta}}(x) - r_{\hat{\theta}}(x, y)))] - \mathbb{E}_{Y \sim \bar{\pi}_{\boldsymbol{\beta}, \mathrm{ref}}(\cdot|x)}[\log(\gamma(\tilde{Z}_{\theta^\star}(x) - r_{\theta^\star}(x, y)))]}{\gamma}$$

$$\leq \frac{\left|\mathbb{E}_{Y \sim \bar{\pi}_{\boldsymbol{\beta}, \mathrm{ref}}(\cdot|x)}[\log(\gamma(\tilde{Z}_{\hat{\theta}}(x) - r_{\hat{\theta}}(x, y)))]\right| + \left|\mathbb{E}_{Y \sim \bar{\pi}_{\boldsymbol{\beta}, \mathrm{ref}}(\cdot|x)}[\log(\gamma(\tilde{Z}_{\theta^\star}(x) - r_{\theta^\star}(x, y)))]\right|}{\gamma}$$

$$\leq \frac{\max\left(|\log(C_{\varepsilon, \mathrm{fkl}})|, \log(\gamma R_{\max} + 1)\right)}{\gamma}, \tag{61}$$

where the last inequality follows from Lemma E.3. The final result holds by combining Theorem 6.2 with (61). $\square$

# F Extension to DPO

As discussed in [Song et al., 2024], DPO can not guarantee any performance under some conditions. In particular, The reverse KL-regularized case can fail under partial coverage conditions, necessitating the Global Coverage Assumption (Assumption 4.3). The forward KL-regularized case requires an even stronger condition: the ratio of reference to policy must be bounded from below away from zero. Specifically, we should have $0 < \inf_{(x,y),\rho(x)>0} \frac{\pi_{\boldsymbol{\beta},\mathrm{ref}}(y|x)}{\pi_\theta(y|x)}$ which is a stronger assumption. For this purpose, we consider the implicit bounded reward assumptions.

Our theoretical results for reverse KL-regularized RLHF and forward KL-regularized RLHF can be applied DPO problems (22) and (23) under the following assumptions.

**Assumption F.1** ((Bounded implicit RKL reward). *For all $y^w, y^l \in \mathcal{Y}$ and $x \in \mathcal{X}$, there exists a constant $B_{\max}$ such that,*

$$\left| \frac{1}{\gamma} \log(\frac{\pi_\theta(y^w|x)}{\pi_{\boldsymbol{\alpha},\mathrm{ref}}(y^w|x)}) - \frac{1}{\gamma} \log(\frac{\pi_\theta(y^l|x)}{\pi_{\boldsymbol{\alpha},\mathrm{ref}}(y^l|x)}) \right| \le B_{\max}. \tag{62}$$

**Assumption F.2** ((Bounded implicit FKL reward). *For all $y^w, y^l \in \mathcal{Y}$ and $x \in \mathcal{X}$, there exists a constant $D_{\max}$ such that,*

$$\left| \frac{1}{\gamma} \frac{\pi_{\boldsymbol{\beta},\mathrm{ref}}(y_i^l|x_i)}{\pi_\theta(y_i^l|x_i)} - \frac{1}{\gamma} \frac{\pi_{\boldsymbol{\beta},\mathrm{ref}}(y_i^w|x_i)}{\pi_\theta(y_i^w|x_i)} \right| \le D_{\max}. \tag{63}$$

**Lemma F.3** (Lemma E.5 from [Huang et al., 2024]). *Under Assumptions 4.1, F.1 and 4.2, we have with probability at least $1 - \delta$ that*

$$\mathbb{E}_{Y^l, Y^w \sim \pi_{\mathrm{ref}}, \pi_{\mathrm{ref}}} \left[ \left( r_{\theta^\star}(x, Y^l) - r_{\theta^\star}(x, Y^w) - r_{\widehat{\theta}}(x, Y^l) + r_{\widehat{\theta}}(x, Y^w) \right)^2 \right]$$

$$\le \frac{128 B_{\max}^2 \exp(4R_{\max}) \log(|\mathcal{R}|/\delta)}{n}. \tag{64}$$

The same results also holds under Assumption F.2.

---

**Theorem F.4.** *Under Assumptions F.1, 4.1, 4.2 and 4.4, there exists constant $C > 0$ such that the following upper bound holds on the optimality gap of DPO based on reverse KL-regularization with probability at least $(1 - \delta)$ for $\delta \in (0, 1/2)$,*

$$\mathcal{J}(\pi_{\theta^\star}^\gamma(\cdot|x), \pi_{\widehat{\theta}}^\gamma(\cdot|x)) \le \gamma C_{\boldsymbol{\alpha}, \varepsilon_{\mathrm{rkl}}} 128 e^{4R_{\max}} B_{\max}^2 \frac{\log(|\mathcal{R}|/\delta)}{n}$$

$$+ C 8 B_{\max} e^{2R_{\max}} \sqrt{\frac{2 C_{\boldsymbol{\alpha}, \varepsilon_{\mathrm{rkl}}} \log(|\mathcal{R}|/\delta)}{n}}.$$

---

*Proof.* The proof is similar to Theorem 5.3 using Lemma F.3. $\qquad\square$

---

**Theorem F.5.** *Under Assumptions F.2, 4.1, 4.2 and 4.4, the following upper bound holds on optimality gap of DPO based on forward KL-regularization with probability at least $(1 - \delta)$ for $\delta \in (0, 1)$,*

$$\tilde{\mathcal{J}}(\tilde{\pi}_{\theta^\star}^\gamma(\cdot|x), \tilde{\pi}_{\widehat{\theta}}^\gamma(\cdot|x)) \le 16 C_{\boldsymbol{\beta}, \varepsilon_{\mathrm{fkl}}} e^{2R_{\max}} D_{\max} \sqrt{\frac{\log(|\mathcal{R}|/\delta)}{n}}$$

$$+ \frac{\max\left( |\log(C_{\varepsilon, \mathrm{fkl}})|, \log(\gamma R_{\max} + 1) \right)}{\gamma}.$$

---

*Proof.* The proof is similar to Theorem 6.3 by using Lemma F.3. $\qquad\square$

# G   Further Discussion

**Coverage Assumption Discussion:** The coverage assumptions for multiple references can differ from the single reference scenario. For the reverse KL-regularized case with reference policy $\widehat{\pi}_{\boldsymbol{\alpha},\mathrm{ref}}(\cdot|x)$, we have:

$$\frac{\pi(y|x)}{\widehat{\pi}_{\boldsymbol{\alpha},\mathrm{ref}}(\cdot|x)} = F_{\boldsymbol{\alpha}}(x) \prod_{i=1}^{K} \left( \frac{\pi(y|x)}{\pi_{\mathrm{ref},i}(y|x)} \right)^{\alpha_i}, \tag{65}$$

where $F_{\boldsymbol{\alpha}}(x)$ is defined in (15). Therefore, we have $\prod_{i=1}^{K} C_{\mathrm{ref},i}^{\alpha_i}$ as the global coverage assumption, where $C_{\mathrm{ref},i} < \infty$ is the global coverage with respect to the $i$-th reference. Note that, using Hölder's inequality, we can show that $F_{\boldsymbol{\alpha}}(x) \leq 1$. A similar discussion applies to the forward KL-regularization scenario with reference policy $\bar{\pi}_{\boldsymbol{\beta},\mathrm{ref}}(y|x)$. Regarding the local reverse KL-ball coverage assumptions (Assumption 4.4), as $\widehat{\pi}_{\boldsymbol{\alpha},\mathrm{ref}}(\cdot|x)$ is defined on common support among all reference models, then the set of policies with bounded

$$\mathbb{E}_{x\sim\rho}[\mathrm{KL}(\pi(\cdot|x)\|\widehat{\pi}_{\boldsymbol{\alpha},\mathrm{ref}}(\cdot|x))] \leq \varepsilon_{\boldsymbol{\alpha},\mathrm{rkl}}, \tag{66}$$

is smaller than each reference model separately. Similarly to global coverage, we can assume that $C_{\boldsymbol{\alpha},\varepsilon_{\mathrm{rkl}}} = \prod_{i=1}^{K} C_{\mathrm{ref},i,\varepsilon_{\mathrm{rkl}}}^{\alpha_i}$.

**Comparison of RKL with FKL:** The RKL and FKL exhibit fundamentally different characteristics in their optimization behavior. RKL between the reference model and target policy, defined as $\mathbb{E}_{\pi_{\theta^\star}}[\log(\pi_{\theta^\star}/\pi_{\mathrm{ref}})]$, demonstrates mode-seeking behavior during optimization. When $\pi_{\theta^\star}$ represents the output policy of RLHF for language model alignment, it may assign zero probability to regions where $\pi_{\mathrm{ref}}$ is positive. Conversely, FKL, expressed as $\mathbb{E}_{\pi_{\mathrm{ref}}}[\log(\pi_{\mathrm{ref}}/\pi_{\theta^\star})]$, exhibits mass-covering properties. Its mathematical formulation requires $\pi_{\theta^\star}$ to maintain non-zero probability wherever $\pi_{\mathrm{ref}}$ is positive. This constraint naturally leads FKL to produce distributions that cover the full support of the reference model, thereby promoting diverse outputs.

**Reference policy in multiple reference model scenario under FKL and RKL:** In the multiple reference model setting, the generalized escort distribution under reverse KL-regularization covers the intersection of the supports of all reference models. Specifically, responses receive zero probability if they lack positive probability in any single reference model. This leads the generalized escort distribution to assign non-zero probabilities only to responses supported by all reference models simultaneously. In contrast, when using the average distribution as the reference model in the forward KL scenario, the resulting distribution covers the union of supports across all reference models, encompassing a broader range of possible responses.

# H    Experiment Details

Implementation code is provided at https://github.com/idanshen/multi_ref.

To ensure fair comparison across algorithms, we began by conducting an independent hyperparameter search for each method. For the GRPO experiment, we explored learning rates of $\{1e\text{-}3, 1e\text{-}4, 1e\text{-}5\}$ and KL coefficients of $\{0.05, 0.1, 0.2\}$. For the DPO experiments, we explored learning rates of $\{1e\text{-}6, 1e\text{-}7, 1e\text{-}8\}$. We also tried different $\gamma$ values but found that the default one works the best in all cases. After selecting the best configuration for each algorithm, we trained each setup three times with different random seeds to estimate variability and compute confidence intervals.

In the case of GRPO, using the full FKL objective would require sampling from the reference model, which roughly doubles training time. To reduce this cost, we instead approximated the FKL term by sampling from the trained model and computing a per-token objective—striking a balance between efficiency and fidelity to the theoretical objective.

Our data splits were chosen to reflect standard practice where possible. For GSM8K, we used the official train-test split. Since UltraFeedback does not provide an official split, we randomly withheld 10% of the dataset and used the corresponding prompts for evaluation.

All experiments were conducted on A100 GPUs. Offline RLHF training used a single GPU, while online training required two. Although multi-reference RL introduces some additional computational requirements—specifically, evaluating logits from another policy—the cost is modest. In offline settings such as DPO, reference model logits can be precomputed and stored, avoiding memory overhead during training. In online settings like GRPO, the reference policy must reside in memory, but placing it on a separate GPU resulted in only a 10% slowdown.

