# OpenReview forum: "KL-Regularized RLHF with Multiple Reference Models: Exact Solutions and Sample Complexity"
_NeurIPS.cc/2025/Conference — NeurIPS 2025 poster_

### Official Review · Reviewer_Gq4o · 2025-06-29

**Clarity:** 3
**Significance:** 3
**Originality:** 2
**Rating:** 4
**Confidence:** 3

**Summary:**

This paper studies RLHF with multiple reference models. Traditionally, RLHF is trained with a single reference model. But that is over restrictive because the model can inherits biases of that single pre-trained source. On the other hand, it is also inefficient because there are a lot of pre-trained models available. So far, there is a gap in understanding this new training paradigm because unlike RLHF with single reference model, there’s no exact solution yet. This paper pushes in this direction and provide solutions for RLHF with multiple reference models for both reserver and forward KL constraint.

**Questions:**

- Line 113-114: It’s also possible that two outputs are sampled from a distribution different from the reference policy? An example of which would be the Antropic HH dataset which is collected offline. And how are outputs sampled with multiple reference polices?
- How realistic is assumption 4.2? Typically reward function are trained based on existing LLMs. For example, one could train the last linear layer of a pre-trained LLM. In that case, the VC dimension is finite but the reward function class is not finite.
- I am always a bit confused about the practicality of coverage assumption. It seems like because outputs are sampled from reference policy, I don’t need lower bound of all \pi_ref(y|x) but just the ones that can be realistically sampled? Although I do acknowledge the technical difficulties involved.
- If the assumptions in 4.2 is not needed for 5.1, maybe move the placement of assumptions?
- In the proof of theorem 5.1, you mentioned that “Clearly, the right side of (29) is maximized when the KL divergence is set to zero.” Is it possible that there exists a policy with non-zero KL divergence but high exponential value? (I'll update my score based on answer to this question since I might have missed something obvious)
- Does theorem 5.1 only works with optimal reward?
- I am confused about what the actual algorithm is? How does it rely on the exact solution of RLHF with multiple reference models? The experiment section is written with a lot of details missing.

**Ethical Concerns:**

["NO or VERY MINOR ethics concerns only"]

**Final Justification:**

I was confused about a technical detail and the confusion is cleared.

**Limitations:**

- I have questions about the limitations of assumptions which are discussed in the question section.

**Quality:**

3

**Strengths And Weaknesses:**

- Strengthes
    - Expect for the experimental section, I think the paper is written very clearly with solid motivation and the theory sections are easy to understand.
    - The research direction on RLHF with multiple reference models is an interesting direction and this paper provides some theoretical justifications in that direction.
- Weaknesses
    - The experimental section can be expanded a bit. I am confused about what algorithm is introduced based on the theoretical results.

---

> ### Author Rebuttal · Authors · 2025-07-29
>
> We thank the reviewer for their comments and generally positive assessment of the paper. We will address their concerns in detail below.
>
> ----
>
> > **In the proof of theorem 5.1, you mentioned that “Clearly, the right side of (29) is maximized when the KL divergence is set to zero.” Is it possible that there exists a policy with non-zero KL divergence but high exponential value? (I'll update my score based on answer to this question since I might have missed something obvious)**
>
> **R1:** Thank you for this question. Note that the second term in equation (29), i.e., the logarithm term, is independent of $\\pi(\\cdot|x)$. In particular, the logarithm term is just dependent on reference policies. As the KL divergence is a non-negative term, the right side of equation (29) is maximized when the KL divergence is set to zero. Therefore, the final solution Eq.(25) holds. For more clarity, we will add this explanation in the proof of Theorem 5.1 in the final version.
>
> $$
>  \\begin{aligned}
> &\\underset{Y\\sim\\pi\\big(\\cdot|x\\big)}{{\\mathbb{E}}}\\big\[r\_{\\theta^{\\star}}\\big(x,Y\\big)\\big\]-\\frac{1}{\\gamma}\\Big(\\sum\_{i=1}^{K}\\alpha\_{i} \\mathrm{KL}\\big(\\pi (\\cdot | x)\\| \\pi\_{\\mathrm{ref},i}(\\cdot | x)\\big)\\Big) \\\\& \= \\frac{1}{\\gamma}\\left(-\\mathrm{KL}(\\pi(\\cdot|x)\\|\\pi\_{\\theta^*}^\\gamma\\big(y|x\\big))+\\log \\left(\\sum\_y \\prod\_{i=1}^K \\pi\_{\\mathrm{ref},i}^{\\alpha\_i}(y|x)\\exp\\left(\\gamma r(x,y) \\right)\\right) \\right),   \qquad\qquad \text{Eq.}(29)
>  \\end{aligned}
> $$
>
>
> > **Line 113-114: It’s also possible that two outputs are sampled from a distribution different from the reference policy? An example of which would be the Antropic HH dataset which is collected offline. And how are outputs sampled with multiple reference polices?**
>
> **R2:** Thanks for raising this point. Sampling preference datasets from the reference policy is a common assumption in theoretical analysis of RLHF and DPO \[1,2,3,4\]. Offline datasets (e.g., Anthropic HH) are valid so long as the behavior policy (the underlying policy for the preference datasets) covers the reference policy in KL divergence. We clarify this point in the final manuscript.
>
> > **How realistic is assumption 4.2? Typically reward function are trained based on existing LLMs. For example, one could train the last linear layer of a pre-trained LLM. In that case, the VC-dimension is finite but the reward function class is not finite.**
>
> **R3:** Assumption 4.2 (“finite reward class”) is a technical assumption for our regret analysis.** Our proofs only use finiteness to obtain $\\log(|\\mathcal{R}|)$ factor in Lemma B.1 and theorems; the results can be extended to **classes with finite capacity** (e.g., finite covering number or pseudo/VC dimension) as discussed in Appendix C. It can also be applied to **a learned linear head on fixed LLM features (pre-trained LLM).** The rates in our bounds are unchanged; only the $\\log(|\\mathcal{R}|)$ term is replaced by the corresponding capacity term. We already discussed to this relaxation in Appendix C and will make it explicit in the final manuscript.
>
> > **I am always a bit confused about the practicality of coverage assumption. It seems like because outputs are sampled from reference policy, I don’t need lower bound of all $\\pi\_{ref}(y|x)$ but just
> the ones that can be realistically sampled? Although I do acknowledge the technical difficulties involved.**
>
> **R4:** The coverage assumption is a uniform assumption which is needed for regret analysis. It is also assumed in \[2,3\]. Note that under integral over all samples, we apply this assumption. Therefore, we need to assume a lower bound on $\\pi\_{ref}(y|x)$ for all $y$.
>
> > **If the assumptions in 4.2 is not needed for 5.1, maybe move the placement of assumptions?**
>
> **R5:** Thank you for your suggestion. We also mentioned in Lines 208-209 that “Note that this result does not rely on the assumptions stated in Subsection 4.2 and in fact holds in greater generality.” We clarified this point in the final manuscript.
>
>
> > **Does theorem 5.1 only works with optimal reward?**
>
> **R6:** Thanks for your great suggestion. Our result in Theorem 5.1 is general and can work with any reward function. We clarify this point in the final manuscript.
>
> > **I am confused about what the actual algorithm is? How does it rely on the exact solution of RLHF with multiple reference models? The experiment section is written with a lot of details missing. I am confused about what algorithm is introduced based on the theoretical results.**
>
> **R7:**  Building on **Theorems 5.1** and **6.1**, we introduce two algorithms:
>
> - **Algorithm 1 — RKL**
>   Replaces the single reference policy with the **geometric mean** of all reference policies for RLHF.
>   *See Eq.&nbsp;60 in **Appendix F** for the explicit DPO loss.*
>
> - **Algorithm 2 — FKL**
>   Uses the **arithmetic mean** of the reference policies, in line with Theorem 6.1.
>
> *The following table* (below) collects every objective used in our experiments. We acknowledge the inconvenience of having them spread all over the main paper due to page limits of the main submission, and we will clarify this discussion in the final version.
>
> | Experiment                                   | Equation                              |
> |----------------------------------------------|---------------------------------------|
> | DPO with multi-reference                     | Equation&nbsp;60 (Appendix&nbsp;F)    |
> | FKL DPO with multi-reference                 | Equation&nbsp;61 (Appendix&nbsp;F)    |
> | On‑policy RL with multi-reference            | Equation&nbsp;16                      |
> | On‑policy RL with FKL and multi-reference    | Equation&nbsp;20                      |
>
>
>
> ----
>
> **References:**
>
> \[1\]: Zhao, Heyang, et al. "Sharp analysis for kl-regularized contextual bandits and rlhf."
> \[2\]: Song, Y, et al. “The importance of online data: Understanding preference fine-tuning via coverage.”
> \[3\]: Chang et al. “Dataset reset policy optimization for rlhf”
> \[4\]: Rafailov, Rafael, et al. "Direct preference optimization: Your language model is secretly a reward model."
> \[5\]: Freedman, David A. "On tail probabilities for martingales." *the Annals of Probability* (1975): 100-118.

---

> > ### Comment · Reviewer_Gq4o · 2025-08-05
> >
> > Sorry for the late reply. Regrading 5.1, I think I kind of assume r and $\pi$ are connected? Then what is $r$ if not the parameterized or true reward?

---

> ### Author Response · Authors · 2025-08-05
> **Response**
>
> Thank you for the helpful observation. In our notation in Theorem 5.1, the symbol \\(r\\) always denotes the *ground-truth reward function* \\(r\_{\\theta^\\star}\\).  In the derivation of Theorem 5.1 the optimisation variable is the policy \\(\\pi\\) alone, while \\(r\_{\\theta^\\star}\\) is treated as a fixed mapping \\((x,y)\\mapsto r\_{\\theta^\\star}(x,y)\\).  This is made explicit in Eq.(29), where \\(r\_{\\theta^\\star}\\) appears inside the objective but is **not** varied during the maximization.
>
> Solving that objective yields the Gibbs–Boltzmann policy based on $r\_{\\theta^\\star}(x,y)$,
> \\\[
> \\pi^{\\gamma}\_{\\theta^\\star}(y\\mid x)\\;=\\;
> \\frac{\\widehat{\\pi}\_{\\boldsymbol{\\alpha},\\mathrm{ref}}(y\\\!\\mid\\\!x)\\,
>       \\exp\\\!\\bigl(\\gamma\\,r\_{\\theta^\\star}(x,y)\\bigr)}
>      {\\displaystyle Z(x)},
> \\\]
> shown in Eq.(25).  Here \\(Z(x)\\) is the normalising constant ensuring \\(\\sum\_{y}\\pi^{\\gamma}\_{\\theta^\\star}(y\\mid x)=1\\).
>
> Because \\(r\_{\\theta^\\star}\\) is fixed, there is **no mutual dependence** between the reward and the policy: \\(\\pi^{\\gamma}\_{\\theta^\\star}\\) adapts to \\(r\_{\\theta^\\star}\\) through the KL-regularization, but \\(r\_{\\theta^\\star}\\) itself never depends on \\(\\pi\\).  Consequently, Theorem 5.1 is fully general and remains valid if one substitutes any other reward function (e.g. a learned surrogate \\(r\_{\\hat{\\theta}}\\)) in place of \\(r\_{\\theta^\\star}\\).  We clarify this discussion in the revised manuscript.
>
>
> ---
>
>
> Since you mentioned that you might consider raising your score if the issue with Theorem 5.1 were clarified, we wanted to kindly follow up and check whether there are any remaining concerns we may not have fully addressed. We hope our explanation resolves the point, but we would be happy to further elaborate if needed.

---

> > ### Comment · Reviewer_Gq4o · 2025-08-05
> >
> > Yes. I'll update my score.

---

> ### Author Response · Authors · 2025-08-06
>
> Thank you for your thoughtful and constructive feedback. We have incorporated your suggestions into the final manuscript. We appreciate your revised rating and your support.

---

### Official Review · Reviewer_S55a · 2025-07-02

**Clarity:** 3
**Significance:** 3
**Originality:** 3
**Rating:** 5
**Confidence:** 3

**Summary:**

To address the significant theoretical challenges of integrating multiple reference models into reinforcement learning with human feedback (RLHF) frameworks, this paper provides the exact solution to the multiple reference model problem under reverse KL-regularized RLHF. A comprehensive theoretical framework is proposed, incorporating rigorous statistical analysis and offering theoretical guarantees on sample complexity. The analysis is further extended to forward KL-regularized RLHF, presenting new insights into sample complexity requirements in multiple reference scenarios. The main contributions are as follows:

1. A comprehensive mathematical framework is proposed for reverse KL-regularized RLHF with multiple reference models, providing the exact solution to this problem and calculating the maximum objective value.

2. Theoretical guarantees are established for the proposed multiple reference model setting under reverse KL-regularization.

3. The multiple reference model scenario is also studied under forward KL-regularized RLHF, and its sample complexity is analyzed.

**Questions:**

For detailed comments, please refer to the Weakness section. If the authors could clarify the issues raised therein, I would be happy to consider raising the score.

**Ethical Concerns:**

["NO or VERY MINOR ethics concerns only"]

**Final Justification:**

I appreciate the authors’ thorough follow-up, which resolved my concerns. Accordingly, I have raised my score to 5 to show my support for accepting this paper.

**Limitations:**

Yes

**Paper Formatting Concerns:**

No major formatting issues found.

**Quality:**

3

**Strengths And Weaknesses:**

Strengths:

1. The theoretical analysis is novel, particularly in the authors' innovative and comprehensive investigation of the sub-optimal gap and optimal gap for both forward KL and backward KL regularizations, effectively filling a gap in the existing theoretical literature.
2. The theoretical results are well supported by experimental validation. The authors designed dual-scenario experiments to verify that their closed-form solution outperforms existing approximate methods and single-reference models.

Weaknesses:

1. The theoretical results are only validated on two relatively small models and on specific tasks, without involving larger-scale models. This limitation may affect the generality and rigour of the conclusions.
2. Although the theoretical proofs are clear and rigorous, the paper lacks intuitive explanations of the theoretical results. It would be valuable if the authors could provide more intuitive interpretations or insights into their theoretical findings.

---

> ### Author Rebuttal · Authors · 2025-07-29
>
> We thank the reviewer for their comments, and generally positive assessment of the paper. We will address their concerns as detailed below.
>
> ----
>
> >  **The theoretical results are only validated on two relatively small models and on specific tasks, without involving larger-scale models. This limitation may affect the generality and rigour of the conclusions.**
>
> **R1**: *Experiment with larger model:*  We provide an additional experiment, this time with a larger model - Qwen 2.5 7B. We train this model on the UltraFeedback [cite] dataset, and evaluate the trained model's win rate against the preferred answer in the test split of the dataset using GPT-4o as LLM-as-a-Judge. We follow the standard protocol of first performing SFT on the dataset before the DPO step.
> As a second reference, we used Qwen 2.5 14B Instruct. Due to the time and resource limit, we only trained a single model at $\alpha=0.5$.
>
> | Model / Setting                              | Score |
> |----------------------------------------------|-------|
> | Base model (Qwen 2.5 7B)           | 8.6 % |
> | SFT model                                    | 43.4 % |
> | DPO (single reference – SFT model)           | 56.4 % |
> | DPO (single reference – 14 B model)          | 59.8 % |
> | **Ours (DPO with both references Eq.60)**          | **66.1 %** |
>
> We can observe that using the geometric mean of reference policies as the reference policy in Eq.60, we can achieve better performance with DPO based on both references (Eq.60).
>
> > **Although the theoretical proofs are clear and rigorous, the paper lacks intuitive explanations of the theoretical results. It would be valuable if the authors could provide more intuitive interpretations or insights into their theoretical findings.**
>
> **R2:** Thank you for the helpful comments. In Appendix G, we provide additional discussion of our theoretical results, we plan to move this appendix to main body in final manuscript, including: (i) a detailed examination of the coverage assumption, (ii) a comparison in terms of other relevant parameters, (iii) an analysis of the reference policy in the multiple-reference setting under both FKL and RKL, and (iv) a comparison between RKL and FKL. Furthermore, regarding the sample complexity analysis, we can observe the upper bound on the suboptimal gap of RKL has a sharper convergence rate $O(1/n)$ in comparison with FKL. Regarding coverage assumption, the global coverage requirement for the RKL-learned policy corresponds to the *geometric mean* of the individual coverage levels between each reference model and the learned policy. In contrast, for FKL, the global coverage requirement is characterized by the *arithmetic mean* of the coverage levels. This distinction highlights how the form of the divergence influences the overall support overlap needed for effective learning.

---

> ### Author Response · Authors · 2025-08-04
> **Feedback**
>
> Dear Reviewer S55a,
>
> Thank you for taking the time to review our paper and for your constructive feedback. In our rebuttal, we have carefully addressed each of the concerns raised in the “Weakness” section as you suggested, and we hope our clarifications were helpful.
>
> Since you mentioned you would consider raising your score if the issues were clarified, we wanted to kindly check if there are any remaining concerns that we might not have fully addressed. We truly appreciate your efforts and the valuable feedback you have offered throughout this process. Please let us know if any additional information is needed.
>
> Best regards,
>
> Authors

---

> > ### Comment · Reviewer_S55a · 2025-08-05
> >
> > I thank the authors for their efforts during the rebuttal phase. My concerns have been addressed, and I have updated my score accordingly.

---

> ### Author Response · Authors · 2025-08-06
>
> Thank you for your thoughtful and constructive feedback. We have incorporated your suggestions into the final manuscript. We appreciate your revised score and your support.

---

### Official Review · Reviewer_3gQj · 2025-07-03

**Clarity:** 2
**Significance:** 2
**Originality:** 2
**Rating:** 4
**Confidence:** 1

**Summary:**

This paper derives closed-form solutions and sample-complexity bounds for reverse and forward KL–regularized RLHF with multiple reference policies. The optimal policy is a normalized geometric mean (reverse KL) or weighted arithmetic mean (forward KL) of references, exponentiated by scaled rewards. Additionally, the paper adds GSM8K  and UltraFeedback as experiments.

**Questions:**

* Can one use approximation for reducing cost in the given analysis framework?

**Ethical Concerns:**

["NO or VERY MINOR ethics concerns only"]

**Final Justification:**

The authors have addressed the questions, and I would maintain my score.

**Limitations:**

Yes.

**Paper Formatting Concerns:**

The manuscript feels space-constrained; consider relocating the detailed assumptions to the appendix to streamline the main text and improve readability.

**Quality:**

2

**Strengths And Weaknesses:**

---
* Quality:
    * Strengths: Provides closed-form multi-reference policy (Eq. (14)) with tight $O(1/n)$ and $O(1/\sqrt n)$ bounds under relaxed coverage.
    * Weaknesses: Relies on bounded rewards and finite reference class; coverage constants $C_{\mathrm{rkl}},C_{\mathrm{fkl}}$ are never instantiated.
---

* Clarity:
    * Strengths: Reverse and forward KL derivations are presented in parallel, and the core policy formula is highlighted.
    * Weaknesses: Heavy notation (escort distributions, functional derivatives) and no running example or notation cheat sheet hamper readability.

---

* Significance:
    * Strengths: Replaces global coverage with a local KL-ball condition, still achieving the best known rates; unifies reverse/forward KL analysis.
    * Weaknesses: Practical impact on large LLMs is unclear—combining $K$ models per update may not scale and experiments are limited to small models.

---
* Originality:
    * Strengths: First to derive exact multi-reference RLHF solutions without surrogate bounds, highlighting geometric vs. arithmetic blends.
    * Weaknesses: Omits discussion of links to errors-in-variables or robust regression literatures that also perturb feature distributions.

---

> ### Author Rebuttal · Authors · 2025-07-29
>
> We thank the reviewer for their comments, and generally positive assessment of the paper. We will address their concerns in detail below.
>
> ---
>
> > **Relies on bounded rewards and finite reference class; coverage constants $C\_{\\epsilon\_{rkl}}$ and $C\_{\\epsilon\_{fkl}}$ are never instantiated.**
>
> **R1:** Regarding the bounded reward and finite class assumptions, Assumption 4.1 and 4.2,, in appendix C, we provided the following discussion:
>
> “These Assumptions are common literature are common in the literature \[1,2,3\]. In particular, Assumption 4.1
> is primarily to enable the use of concentration inequalities like Freedman’s inequality, which require bounded differences (as in Lemma B.1). However, this assumption can be relaxed under certain growth conditions, as discussed in \[5\]. Moreover, even when the original reward function is unbounded or sub-Gaussian—as is often the case in human preference modeling—it is possible to apply a monotonic, bounded transformation to the rewards. For instance, one can use the cumulative distribution function (CDF) of the reward under a reference model to normalize the rewards into a bounded range, as proposed in \[5\]. This approach also retains the essential ordering of preferences and supports handling sub-Gaussian behavior in the transformed space. Regarding finite class, we can apply covering number and relax this assumption as utilized in \[1\].”
>
> Regarding coverage constants $C\_{\\epsilon\_{rkl}}$ and $C\_{\\epsilon\_{fkl}}$, we also utilized these constants in our regret analysis. These assumptions help us to provide more refined coverage assumptions due to regularization which keeps the final policy close to the reference policy.
>
>
> > **Heavy notation (escort distributions, functional derivatives) and no running example or notation cheat sheet hamper readability.**
>
> **R2:** Thanks for your suggestion. We add a table of notations in the appendix for helping readers to follow the notations.
>
> >  **Practical impact on large LLMs is unclear—combining $K$ models per update may not scale and experiments are limited to small models.**
>
> **R3:**  Thanks for the question about scaling. Our method does not fuse full models at each update. We only ask the reference models to score the text; the main model does the learning. In offline algorithms (like DPO), those scores can be saved once and reused, so training time stays the same. In online runs with two references, the extra time we saw was small (about 10%), as generations and backpropagation are the steps that require most of the time during training.
>
> > **Omits discussion of links to errors-in-variables or robust regression literatures that also perturb feature distributions.**
>
> **R4:** Thank you for raising this point. We agree it is helpful to situate our analysis relative to errors‑in‑variables (EIV) and robust regression, both of which study **perturbations of the data‑generating process**. Our work perturbs a different object: the **policy over outputs** (responses) and constrains it to stay near an **ensemble of reference policies** via KL. In particular, our reverse‑KL objective with K references is equivalent to a single KL to the *normalized geometric mean* (“generalized escort”) of the references, (Thm. 5.1), whereas the forward‑KL objective reduces to a KL from the *arithmetic mean* of the references, (Thm. 6.1). This is closer to **distributionally robust optimization (DRO)** with f‑divergence ambiguity sets than to classical EIV: we regularize the *candidate policy distribution* relative to a baseline (reference ensemble), rather than correcting parameter estimates under noisy covariates. We will add a paragraph clarifying these links, citing classic EIV/measurement‑error references (e.g., Fuller \[6\]) and robust regression (Huber \[7\]), and noting the DRO connection (KL ambiguity sets).
>
> > **Can one use approximation for reducing cost in the given analysis framework?**
>
> **R5:** Thank you for your insightful comment.
>
> - **Inference cost.** The additional cost of using multiple reference policies can be mitigated through **speculative decoding**, which reuses partial computations to keep overhead low.
> - **Alternative cost metrics.** If you have a different cost model in mind, please let us know and we will be happy to address it in detail.
>
> > **The manuscript feels space-constrained; consider relocating the detailed assumptions to the appendix to streamline the main text and improve readability.**
>
> **R6:** Thanks for your suggestion. We consider it for the final version of the manuscript.
>
> ----
>
> **References:**
>
> \[1\]: Zhao, Heyang, et al. "Sharp analysis for kl-regularized contextual bandits and rlhf."
> \[2\]: Song, Y, et al. “The importance of online data: Understanding preference fine-tuning via coverage.”
> \[3\]: Chang et al. “Dataset reset policy optimization for rlhf”
> \[4\]: Freedman, David A. "On tail probabilities for martingales." *the Annals of Probability* (1975): 100-118.
> \[5\]: Balashankar, Ananth, et al. "InfAlign: Inference-aware language model alignment."
> \[6\]: Fuller, Wayne A. *Measurement error models*. John Wiley & Sons, 2009\.
> \[7\]: Huber, Peter J. "Robust regression: asymptotics, conjectures and Monte Carlo." *The annals of statistics* (1973): 799-821.

---

### Official Review · Reviewer_vZWY · 2025-07-03

**Clarity:** 3
**Significance:** 2
**Originality:** 3
**Rating:** 4
**Confidence:** 2

**Summary:**

This paper provides a theoretical analysis of KL-regularized RLHF using multiple reference models. The authors derive the first exact closed-form solutions for both reverse KL (RKL) and forward KL (FKL) regularized objectives in this multi-reference setting. They further provide sample complexity guarantees for the optimality and sub-optimality gaps for both frameworks.

**Questions:**

1. The normalization constant $F_\alpha(x)$ in Eq. 15 requires a sum over the entire vocabulary, which is often intractable. How was this handled in the offline DPO experiment? Does the corresponding DPO loss function for multiple references (presumably in App. F) circumvent this explicit calculation?
2. The coverage constants $C_\{ε_\{rkl}}$ are central to the bounds but their dependency on the number (K) and diversity of reference models is unclear. Cud you provide some intuition on how these constants might scale with K? Does the bound become vacuous if the reference models are very different?
3. The empirical gains from using multiple references ($\alpha>0$) are positive but appear somewhat marginal in Fig 1. In what practical scenarios do you hypothesize the multi-reference approach would provide a much more substantial improvement over a strong single-reference baseline?

**Ethical Concerns:**

["NO or VERY MINOR ethics concerns only"]

**Limitations:**

Yes

**Paper Formatting Concerns:**

Yes

**Quality:**

2

**Strengths And Weaknesses:**

**Strengths:**
*   **Novel Theoretical Contribution:** The core strength is the derivation of an exact closed-form solution for the multi-reference RKL-RLHF problem (Thm 5.1). This is a significant step beyond prior work (Le et al., 2024) which relied on a lower-bound approximation. The resulting optimal policy as a normalized geometric mean of reference policies (Eq. 15) is an elegant result.
*   **Comprehensive Analysis:** The anlysis is thorough, covering both RKL and FKL regularization. For each, the paper derives optimality and sub-optimality gap bounds (Thms 5.2, 5.3, 6.2, 6.3), providing a complete theoretical picture. The comparison in Table 1 clearly situates the contributions within existing literature.
*   **Strong Motivation:** The paper makes a compelling case for using multiple reference models to improve diversity, reduce bias, and leverage the growing ecosystem of open-source LLMs. Teh motivation is well-grounded and timely.

**Weaknesses:**
*   **Limited Empirical Validation:** This is the main weakness. While the experiments in Sec 8 support the theory, they are limited in scope. The use of relatively small models (0.5B) and only two datasets (GSM8K, UltraFeedback) makes it hard to gauge the practical significance for SOTA LLM alignment. The performance gains shown in Fig 1 over the single-reference baseline are modest.
*   **Practicality of Assumptions and Implementation:** The paper is heavily theoretical. The practical implications of the Local KL-ball Coverage assumption (4.4, 4.5) are not fully explored. For instance, how does the constant $C_\{ε_\{rkl}}$ scale with the number of reference models? Also, the computation of the normalization term $F_\alpha(x)$ in Eq. 15 seems computationally prohibitive (sum over all $y$ in $Y$); more discussion on how this is handled in the DPO-style experiment would be valuable in the main text.
*   **Significance of FKL Results:** While the FKL analysis is complete, the experimental results (Fig 1a, 1b) show it consistently underperforming the RKL approach. This raises questions about teh practical utility of the multi-reference FKL methd, even if the theory is sound.

---

> ### Author Rebuttal · Authors · 2025-07-29
>
> We thank the reviewer for their comments and generally positive assessment of the paper. We will address their concerns in detail below.
>
> ----
>
> > **Limited Empirical Validation: This is the main weakness. While the experiments in Sec 8 support the theory, they are limited in scope. The use of relatively small models (0.5B) and only two datasets (GSM8K, UltraFeedback) makes it hard to gauge the practical significance for SOTA LLM alignment. The performance gains shown in Fig 1 over the single-reference baseline are modest. The empirical gains from using multiple references ( $\\alpha\>0$) are positive but appear somewhat marginal in Fig 1\. In what practical scenarios do you hypothesize the multi-reference approach would provide a much more substantial improvement over a strong single-reference baseline?**
>
> **R1:** Regarding the performance gains shown in Fig. 1, we politely disagree that a gain of 30% in success rate in GSM8K (from ~30% of the single-reference policy to ~60% of the multi-reference policy) is “marginal”. We recognize the reviewer’s worry that we only used small models and therefore we tried larger models.
>
> *Experiment with larger model:*  We provide an additional experiment, this time with a larger model - Qwen 2.5 7B. We train this model on the UltraFeedback [cite] dataset, and evaluate the trained model's win rate against the preferred answer in the test split of the dataset using GPT-4o as LLM-as-a-Judge. We follow the standard protocol of first performing SFT on the dataset before the DPO step.
> As a second reference, we used Qwen 2.5 14B Instruct. Due to the time and resource limit, we only trained a single model at $\alpha=0.5$.
>
> | Model / Setting                              | Score |
> |----------------------------------------------|-------|
> | Base model (Qwen 2.5 7B)           | 8.6 % |
> | SFT model                                    | 43.4 % |
> | DPO (single reference – SFT model)           | 56.4 % |
> | DPO (single reference – 14 B model)          | 59.8 % |
> | **Ours (DPO with both references Eq.60)**          | **66.1 %** |
>
> We can observe that using the geometric mean of reference policies as the reference policy in Eq.60, we can achieve better performance with DPO based on both references (Eq.60).
>
> > **The normalization constant $F\_{\\alpha}(x)$  in Eq. 15 requires a sum over the entire vocabulary, which is often intractable. How was this handled in the offline DPO experiment? Does the corresponding DPO loss function for multiple references (presumably in App. F) circumvent this explicit calculation?**
>
> **R2:**  We thank the reviewer for his question. The DPO loss does not need the normalization $F\_{\\alpha}(x)$ factor to be explicitly calculated, and therefore, the nominator of equation 15 can be easily calculated using the per-token probability of the reference models. In particular, in DPO experiments, as shown in loss function of DPO for both RKL and FKL, the normalization constants are removed. Therefore, this constant is handled in DPO experiments.
>
>
> > **The coverage constants  $C\_{\\alpha,\\epsilon\_{rKL}}$  are central to the bounds but their dependency on the number (K) and diversity of reference models is unclear. Could you provide some intuition on how these
> constants might scale with K? Does the bound become vacuous if the reference models are very different?**
>
> **R3**: Thanks for your point. Regarding the constant \\( C\_{\\alpha, \\epsilon\_{\\text{RKL}}} \\), we have shown that
> \\\[
> C\_{\\epsilon\_{\\text{RKL}}} \= \\prod\_{i=1}^K C\_{\\text{ref}, i, \\epsilon\_{\\text{RKL}}}^{\\alpha\_i}.
> \\\]
> Therefore, \\( C\_{\\alpha, \\epsilon\_{\\text{RKL}}} \\) corresponds to the geometric mean of the per-reference coverage constants \\( C\_{\\text{ref}, i, \\epsilon\_{\\text{RKL}}} \\), weighted by \\( \\{\\alpha\_i\\}\_{i=1}^K \\). To have a non-vacuous upper bound, we need to have \\( C\_{\\text{ref}, i, \\epsilon\_{\\text{RKL}}}^{\\alpha\_i} > 0 \\) for all $i$. It means that we should have a coverage of each single model policy with respect to the reference model.
>
>
> > **Significance of FKL Results: While the FKL analysis is complete, the experimental results (Fig 1a, 1b) show it consistently underperforming the RKL approach. This raises questions about the practical utility of the multi-reference FKL method, even if the theory is sound.**
>
> **R4:** RLHF with FKL regularization was originally proposed by \[1\], but its regret analysis has been overlooked. In our work, we address this gap by providing a theoretical analysis of FKL. For the setting with multiple reference models, we have the final reference policy as the arithmetic mean of the individual reference policies for FKL regularization. Interestingly, we observe that using the geometric mean can lead to better performance than the arithmetic mean. Furthermore, as mentioned in Appendix H, in the case of GRPO, using the full FKL objective would require sampling from the reference model, which roughly doubles training time. To reduce this cost, we instead approximated the FKL term by sampling from the trained model and computing a per-token objective—striking a balance between efficiency and fidelity to the theoretical objective.
>
> ---
>
> **References:**
>
> \[1\]: Wang, Chaoqi, et al. "Beyond Reverse KL: Generalizing Direct Preference Optimization with Diverse Divergence Constraints.".

---

### Note · Authors · 2025-08-12

We thank the AC and reviewers for the careful evaluation and constructive discussion. Our goal in this work is to clarify the theory of RLHF with **multiple** reference policies and make it usable in practice. As final remark, we would like to summarise the main points of our rebuttal as follows.

**Novelty.** We give the first exact closed-form solution for the multi-reference **reverse-KL** objective (the target policy is proportional to the geometric mean of references times an exponentiated reward), and a parallel treatment for **forward-KL** (arithmetic mean). We accompany both with finite-sample guarantees under a *local KL-ball* coverage assumption; the RKL bounds are sharper than FKL (RKL achieves a faster rate for sub-optimal gap).

**Practical clarifications.** (i) The partition function/normalizer in the RKL solution is *not* needed in preference-learning objectives: in DPO-style losses it cancels, so we only require per-token probabilities from the references.  (ii) The coverage constants for RKL with multiple references combine **geometrically** (weighted by reference mixing weights);

**Empirical update.** To address scope concerns, we added a larger-model experiment (Qwen 2.5 7B trained on UltraFeedback, LLM-as-a-judge evaluation). Using a second 14B instruct model as an additional reference and the multi-reference DPO derived from our theory, the win rate improved from 59.8% (best single-reference) to 66.1% with both references (single run due to compute). This supports the main claim: aggregating diverse references via our theory can outperform strong single-reference baselines.

**On Theorem 5.1.** The reward is fixed during optimization; the policy is the only variable. This resolves the reviewer’s concern about maximizing the objective: the KL term is minimized at zero, yielding the Gibbs–Boltzmann form; the result also holds if one substitutes a learned surrogate reward.

We thank the AC for considering these final remarks and the reviewers for their thoughtful engagement.

---

### Decision · Program_Chairs · 2025-09-17

**Decision:**

Accept (poster)

**Comment:**

The paper presents a theoretical study on RLHF with multiple reference models. The main contribution of the paper is the novel theoretical analysis that provides the exact solution in the reverse KL-regularized RLHF with multiple reference models. After an effective rebuttal and discussion period, the reviewers are satisfied with the authors feedback and all appreciate the main contribution of the paper. Meanwhile the reviewers also pointed out some issues of the paper, mainly related to insufficient empirical validation of the results. Overall the paper meets the bar of NeurIPS, and thus I would like to recommend acceptance of the paper. The authors should provide a detailed revision addressing all issues raised by the reviewers.

AC